# Modeling land surface processes over a mountainous rainforest in Costa Rica using CLM4.5 and CLM5

Jaeyoung Song[1], Gretchen R. Miller[1], Anthony T. Cahill[1], Luiza Aparecido[2,3], and Georgianne W. Moore[2]

[1]Department of Civil Engineering, Texas A&M University, 3136 TAMU, College Station, TX, 77843
[2]Department of Ecosystem Science and Management, Texas A&M University, 2138 TAMU, College Station, TX, 77843
[3]School of Life Sciences, Arizona State University, Tempe, AZ, 85287

**Correspondence:** Gretchen Miller (gmiller@tamu.edu)

**Abstract.** This study compares the performance of the Community Land Models (CLM4.5 and CLM5) against tower and ground measurements from a tropical montane rainforest in Costa Rica. The study site receives over 4,000 mm of mean annual precipitation and has high daily levels of relative humidity. The measurement tower is equipped with eddy-covariance and vertical profile systems able to measure various micrometeorological variables, particularly in wet and complex terrain. In this
work, results from point-scale simulation for both CLM4.5 and its updated version (CLM5) are compared to observed canopy flux and micro-meteorological data. Both models failed to capture the effects of frequent rainfall events and mountainous topography on the variables of interest (temperatures, leaf wetness, and fluxes). Overall, CLM5 alleviates some errors in CLM4.5 but CLM5 still cannot precisely simulate a number of canopy processes for this forest. Soil, air, and canopy temperatures, as well as leaf wetness, remain too sensitive to incoming solar radiation rates despite updates to the model. As a result, daytime
vapor flux and carbon flux are overestimated, and modeled temperature differences between day and night are higher than those observed. Slope effects appear in the measured average diurnal variations of surface albedo and carbon flux, but CLM5 cannot simulate these features. This study suggests that both CLM models still require further improvements concerning energy partitioning processes, such as leaf wetness process, photosynthesis model, and aerodynamic resistance model for wet and mountainous regions.

## 1 Introduction

Tropical forests play a critical role in determining regional and global climate. Due to their significance for the global water (Zhang et al., 2010; Choudhury and DiGirolamo, 1998) and carbon cycles (Huntingford et al., 2013; Beer et al., 2010), accurate modeling of tropical regions is important for the prediction of future climate and climate change impacts. While tropical forests occupy only 16% of the global land, forests in the tropics house 25% of the carbon stocks found in the terrestrial biosphere, and
account for 33% of global net primary production (NPP) (Bonan, 2008). They account for 33% of terrestrial evapotranspiration (ET), which ranges from 1,000 mm up to 2,200 mm per year and transpiration (TR) occupy its 70% (Schlesinger and Jasechko, 2014; Kume et al., 2011; Fisher et al., 2009; Loescher et al., 2005; Sheil, 2018). Hydrological processes in the humid tropics are also distinctly characterized by warm, uniform temperatures, large inter-annual and spatial variability, intense rainfall, and

greater energy exchange than a temperate forest accelerated by low albedos and high evaporative cooling (Wohl et al., 2012;

Bonan, 2008). The loss of such forests by climate change or human impact can influence their local climate, but also more remote regions (Lawrence and Vandecar, 2014).

Hence, building accurate land-surface models (LSMs) is important. LSMs, as a component of Earth system models (ESMs), simulate the exchange of heat, water vapor, and carbon dioxide between the terrestrial surface and the atmosphere, based essentially on the partitioning of net radiation (Wang et al., 2016). The models have been used for the prediction of future

climate and also its impacts on the land surface such as tropical and extra-tropical forests (Cox et al., 2013; Huntingford et al., 2013).

However, the models do not yet successfully capture the underlying complexity of land-atmosphere interactions (Cai et al., 2014; Wang et al., 2014; Lawrence et al., 2011; Oleson et al., 2010). In particular, LSMs are known to make significant errors in the prediction of carbon and water fluxes for tropical regions. The reasons for these issues are not entirely clear, even

though significant improvements have been made in this field of study (i.e., empirically and mechanistically). Lawrence et al. (2011) compared estimates obtained using two versions of the Community Land Model (CLM3.5 (Oleson et al., 2008) and CLM4.0 (Oleson et al., 2010)) against observed sensible and latent heat flux data from FLUXNET (Baldocchi et al., 2001). They found that CLM4.0 improved predictions compared to CLM3.5 for most sites across the network, but continued to show low agreement for tropical sites. Bonan et al. (2011) updated CLM4.0 by modifying the structure of radiative transfer model

and physiological parameters for canopy processes, which resulted in notable improvements in CLM4.5 (Oleson et al., 2013) but overestimation of carbon and water vapor fluxes persisted in areas closest to the equator. The deficit is especially true for tropical wet mountain rainforests, which have rarely been studied in the context of improving global LSMs, due to the lack of long-term/uniformly distributed measurement and the small number of observation sites (Fisher et al., 2009; Wohl et al., 2012).

To improve land surface models addressing tropical ecosystem biosphere-atmosphere interactions, accurately partitioning net radiation (energy) and water is critical for these models, especially with respect to estimating latent heat flux. Many studies maintain that vapor fluxes in the tropical site are highly correlated ($\approx 87\%$) with net radiation (Andrews, 2016; Fisher et al., 2009; Hasler and Avissar, 2007; Loescher et al., 2005). Others found that leaf wetness is also an important control (Andrews, 2016; Giambelluca et al., 2009). Some studies indicate that the impact of leaf wetness status on the model (which can

contribute 8%-20% of ET) can be detected depending on the canopy water storage capacity and rainfall pattern, although short duration and high intensity rainfall does not significantly affect canopy evaporation (Kume et al., 2011; Loescher et al., 2005). For tropical sites, therefore, the interaction of interception and its evaporation must be included in the modeling framework. Aerodynamic conductance has also been considered as a strong driver for evapotranspiration in tropical forest because the large amount of precipitation and frequently wetted canopy conditions control leaf conductance (Shuttleworth, 1988; Loescher et al.,

2005). Vapor pressure deficit (VPD) has been shown to only slightly influence ($\approx 14\%$ predictor) tropical ET (Fisher et al., 2009; Kume et al., 2011). However, when assessing these studies, we can notice that the studies all highlight the importance and difficulties of quantifying canopy-related water fluxes. ET dynamics are dependent on how these micrometeorological variables are related to the latent heat flux within the energy balance. In tropical forests, the Bowen ratio is consistently less than one

(Loescher et al., 2005), which implies that net radiation highly correlated with latent heat flux. Moreover, the forest canopy acts like a well-watered crop without water limits (Loescher et al., 2005; Hasler and Avissar, 2007; Kume et al., 2011). Hence, how to accurately track water movement within the system (water balance) and predict the ET proportion of net radiation (energy balance) is still a critical question.

Water-related variables are not our only concern, and cannot be independently considered in Earth system or land surface system models. Other energy balance components and physiological elements (e.g., thermal flux, radiative transfers, photosynthesis, and respiration) are likewise important because they are dependent on the water. Normally, all LSMs handle such main variables (e.g., heat/vapor flux, carbon flux, and net radiation). However, recently the modeling community has embraced additional components in order to represent more realistic processes and to resolve research questions related to soil carbon and nitrogen cycling (Thornton et al., 2007), multi-layer plant canopies (Ryder et al., 2016; Launiainen et al., 2015; Bonan et al., 2018), and even more sophisticated systems (e.g., urban settings, heat stress effects) (Lawrence et al., 2018; Buzan et al., 2015). These changes have led to the development of a plethora of sub-models, making it difficult to identify a specific sub-model or set of sub-models from which model error arises.

Land surface models have gradually increased in resolution with the improvement of observations through remote-sensing technology. These changes have highlighted the importance of spatial variability in the land surface system. However, the models still cannot fully reflect the complexity of the surface. The current too simplistic parameterization is one cause of model's error (Singh et al., 2015; Wood et al., 2011). For instance, hydrological processes are well studied at the catchment scale and reflect topographic gradients, but LSMs are known to simplify the effect of the topographic slope (Fan et al., 2019a). Critical zone science has a gap from the Earth system model which normally focuses on vertical flow (Fan et al., 2019a; Clark et al., 2015). The failure to reflect spatial heterogeneity and hydrologic connectivity between large scale process (land-atmosphere fluxes) and microscale process (biogeochemical interactions) can be problematic (Clark et al., 2015).

Hence, in order to properly parametrize global LSMs and to precisely represent complicated systems, such as the tropics and mountains, it is necessary to continue to diagnose land surface models using site-based data. Unique sites like tropical mountain forests are valuable testbeds for model improvement because their environment is an "edge case" for the model; model calibration under more extreme climate conditions can provide valuable insight for the utility of these models under conditions of climate change. Using detailed variables, such as soil moisture/temperature, interception, and stomatal conductivity, site-based studies can identify and alleviate errors in model sub-components. Such errors cannot be easily detected by the analysis of more integrative variables, such as albedo or net radiation.

Measurements including eddy-covariance tower systems have been widely used for the advance of global land surface models via calibration and validation (Bonan et al., 2012; Zaehle and Friend, 2010; Larsen et al., 2016; Chaney et al., 2016). Gridded global data from the FLUXNET network is also available for model development at large scales (Bonan et al., 2011; Jung, 2009). However, point-scale and stand-scale studies still form a core component of research at regional to global scales. In this study, CLM4.5 (Oleson et al., 2013) and its updated version (CLM5) (Lawrence et al., 2018) are employed, and micrometeorological datasets from a tropical rainforest in Costa Rica are compared with these simulation results. The objectives are four-fold:

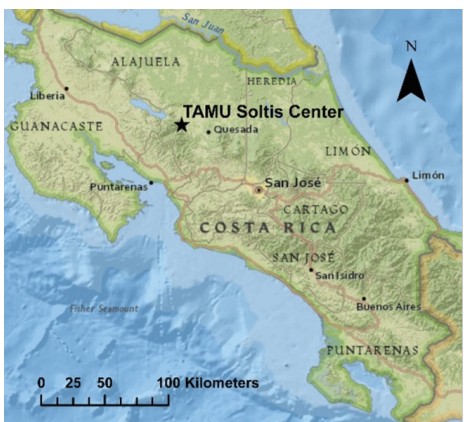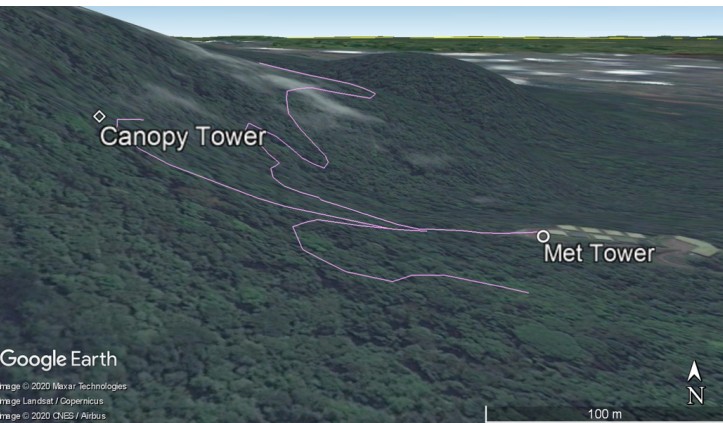

**Figure 1.** The map of the study site and the locations of the two towers (Andrews, 2016).

1. To compare the default mode and point-scale predictions of both CLM 4.5 and CLM 5.0 against micrometeorological and flux measurements collected in a Costa Rican wet montane tropical forest;

2. To identify the improvements in performance between the two CLM versions and shortcomings remaining in the newer version (CLM5);

3. To discern errors caused by the unique environment at our study site (i.e., frequent rainfall and mountainous topography) and to identify too simplistic formulations and incorrect parameters (i.e., interception/leaf-wetness models, photosynthesis models, etc.); and

4. To determine which canopy-atmosphere processes (i.e., sub-models) are most poorly represented, in order to suggest priorities for future model improvements.

## 2 Methodology

### 2.1 Study Site

The field site is located at the Texas A&M University Soltis Center near San Isidro de Peñas Blancas in Costa Rica ($10°23'13''N$, $84°37'33''W$, about 600 m above sea level) [Figure 1]. It shares a boundary with the Children's Eternal Rainforest. This area has a mean annual temperature of 24 $C°$, relative humidity of 85%, and precipitation of 4200 mm (Teale et al., 2014). The study area is classified as a moist, tropical premontane forest. The canopy height ranges from 24 to 45 m, and is located on a steep eastern slope (Aparecido et al., 2016; Jung, 2009). Rainfall is frequent, and a little over two-thirds of days have one or more rain events. The dry season starts from January and continues until April, and the mean rainfall is about 195 mm per month. The wet season is from May until the end of the year: the average rainfall in the wet season is approximately 470 mm per month (Teale et al., 2014; Aparecido et al., 2016).

## 2.2 Micrometeorological measurements

The site has two primary biometeorological measurement locations [Figure 1]. The main weather tower (hereafter called "Met Tower") is located in a flat, grass-covered clearing at the base of the mountain. The walk-up canopy access tower (hereafter called "Canopy Tower") is located within the forest, on the eastern slope. The Met Tower measures meteorological conditions without the influence of canopy processes and structure. Precipitation ($mm$; TE525, Campbell Scientific, Logan, UT), incoming solar radiation, net radiation ($W \cdot m^{-2}$; CNR1, Campbell Scientific), air temperature ($C^\circ$; HMP60, Campbell Scientific), and relative humidity data (%; HMP60, Campbell Scientific) have been collected since 2010. The Canopy Tower has collected the same variables as the Met Tower (with exception of precipitation). A suite of additional measurements, including greenhouse gas concentrations and fluxes, soil moisture, leaf wetness, and sap flow have been collected at the Met Tower since 2014. An infrared, trace-gas profile system (AP200, Campbell Scientific, Logan, UT) and an eddy-covariance system (LI-7200, LI-COR, Lincoln, NE; CSAT3, Campbell Scientific, Logan, UT) are used to collect micrometeorological data at various heights, including concentrations and fluxes of vapor (i.e., $H_2O$) and carbon dioxide (i.e., $CO_2$), wind speed and its direction, and air temperature. Additional data are also collected to track canopy processes: leaf wetness sensors at four different heights (LWS, Decagon Devices, Utah), photosynthetically active radiation (PAR) profiles (LI-190, LI-COR) at five heights, leaf area index (LAI) profile using a lined PAR sensor (LI-191, LI-COR) and Beer-Lambert law (Appendix A), leaf temperature sensors for sunlit and shade leaves (SI-111, Apogee Instruments, Logan, UT), soil heat flux (HFT3, Campbell Scientific), soil temperature (5TE, Decagon Devices, WA), soil moisture (EC-4 and 10HS, Decagon Devices, WA), soil respiration (LI-8100A, LI-COR) and transpiration from sap flow system. Aparecido et al. (2016) and Andrews (2016) present more detailed information about the sap flow system and the profile measurements, respectively. The datasets for this site, from 2014 to 2017, are available via the OAKTrust repository (Miller et al., 2018a, b, c, d).

While the Canopy Tower exceeds the average canopy height, some known interference is present from a nearby emergent tree [Figure 2], leading to a large gap in the canopy in-between heights of roughly 30 and 40 m. This configuration leads to two main challenges. Above the gap, the upslope tree (emergent tree) provides a significant degree of shading, which leads to a 70% reduction in PAR between measurements at the down-slope canopy surface (32 m) and above the emergent tree (44 m). We also note that this configuration makes the eddy-covariance method less than ideal. However, the sonic anemometer and infrared gas analyzer (IRGA) are located at 33 m height, extending away from the tower and clear of obstructions in both the upwind and downslope directions [Figure 2]. As shown in the Figure 1 and 2, predominant winds flow parallel to the valley (N-S) and not perpendicular to this eastern mountain slope. This configuration allows us to capture fluxes, albeit under a narrowed set of ambient conditions. Thus, these data are not necessarily sufficient for recording long-term, integrated measures of ecosystem-level variables, like gross primary production. However, they are suitable for testing and validating models, despite the heterogeneous structure created by the emergent tree.

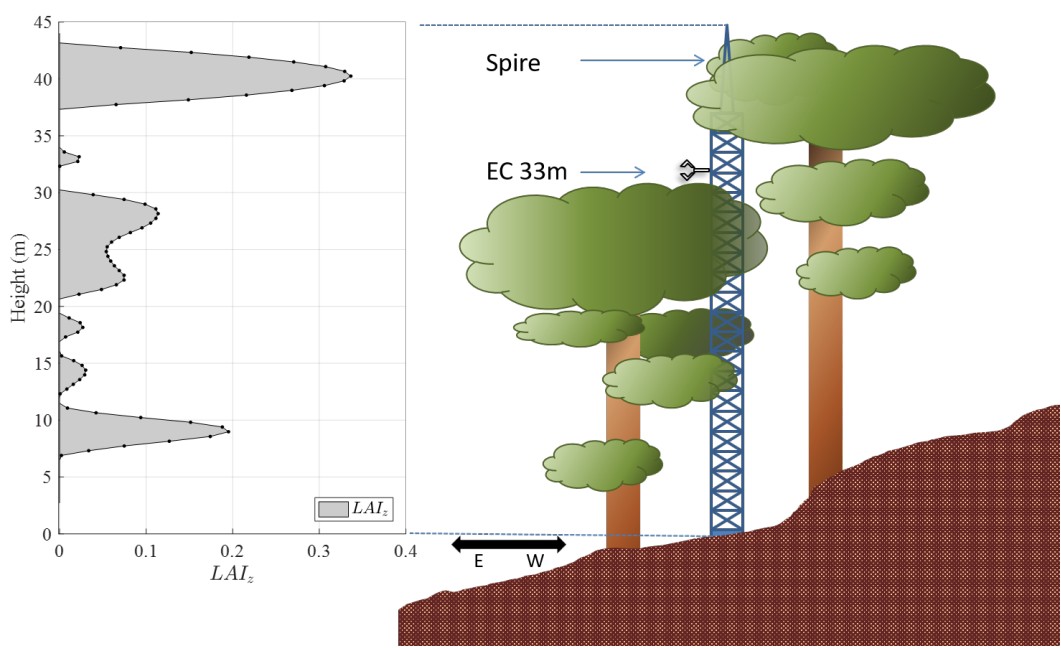

**Figure 2.** Sketch of Canopy Tower located in a plot within a mature premontane moist tropical forest in Costa Rica (right) with LAI profiles highlighted (left) along with the location of the eddy-covariance system (EC, 33 m) and the Spire (44 m), which holds the net radiometer. The leaf area index is given at 22 discrete points (100 points by spline interpolation) in the canopy ($LAI_z$), and its sum ($LAI$) is equal to $6\ m^2 \cdot m^{-2}$ for this stand. The $LAI_z$ was estimated based a light profile data and Beer-Lambert law (Vose et al., 1995). The method for measuring and deriving $LAI_z$ is explained in Appendix A.

## 2.3 Model Description

In this section, we briefly describe CLM's structure and its formulation of the energy balance equation. Given the site's extremely high humidity and annual precipitation, we hypothesize that the sub-models related to water fluxes are the main sources of prediction errors, and as such, the discussion focuses primarily on them. More detailed descriptions can be found in the technical manual (Lawrence et al., 2018; Oleson et al., 2013, 2010). Hereafter we use CLM in a general sense, applicable to both CLM4.5 and CLM5, but provide the specific version number when distinguishing their respective behavior or the effects of
recent code modifications.

   CLM calculates the radiative transfer through the canopy and the ground surface, using the Two-stream approximation method (Dickinson, 1983; Sellers et al., 1992; Bonan, 1996; Oleson et al., 2013), which is a starting point for land surface models to determine the exchange of energy. In the procedure, the canopy structure (e.g., LAI, leaf angle) and the current condition (e.g., wetness, solar angle) are main controllers to determine the absorptivity of incoming solar radiation (albedo).
Based on the absorbed incoming energy, fluxes of sensible heat, latent heat, and soil heat are estimated using the energy balance

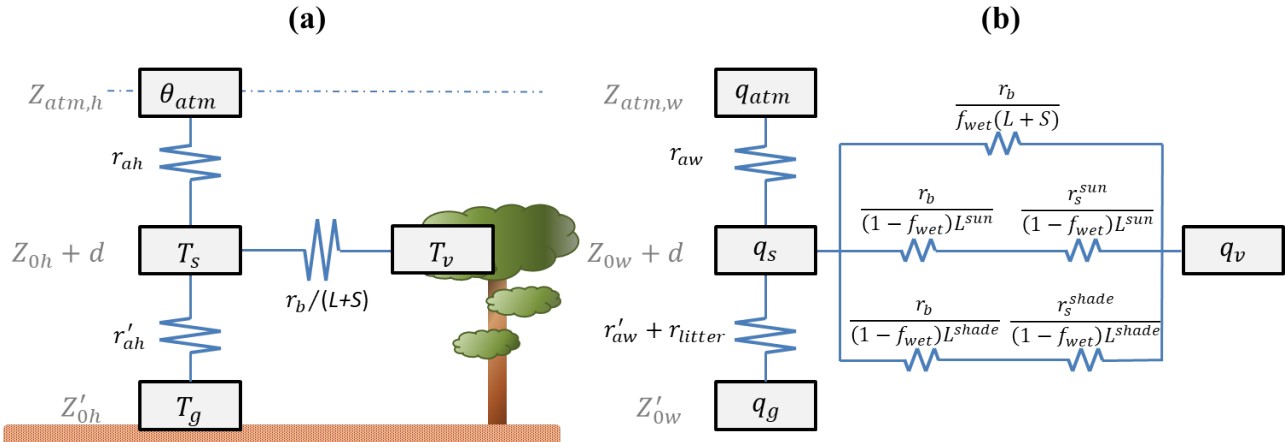

**Figure 3.** Resistance network schemes incorporated within CLM for $(a)$ sensible heat flux and $(b)$ latent heat flux. Main state variables are atmospheric potential temperature ($\theta_{atm}$) and specific humidity ($q_{atm}$), canopy air temperature ($T_s$) and specific humidity ($q_s$), leaf temperature ($T_v$) and its corresponding specific humidity ($q_v$), and ground temperature ($T_g$) and its corresponding specific humidity ($q_g$). Relevant heights are the atmospheric reference height ($z_{atm}$), the canopy roughness heigth ($Z_0$), the groundwater roughness height ($Z_0'$), and the displacement height ($d$). Resistances are specified by their scalar ($h$ for heat and $w$ for water vapor), type ($a$ for aerodynamic, $b$ for boundary layer, $s$ for stomatal, or $litter$ for litter), and lighting ($sun$ or $shade$). Leaf wetness also exerts control on fluxes, via a wetness fraction ($f_{wet}$) and $(L+S)$ is leaf and stem $area$ index. Figure adapted after Oleson et al. (2013).

equation. For example, the canopy energy balance can be written as a function of vegetation temperature ($T_v$):

$$-S_v + L_v(T_v) + H_v(T_v) + LE_v(T_v) = 0 \tag{1}$$

where $S_v$ is the absorbed solar radiation by canopy, $L_v$ is the long wave radiation emitted by canopy, $H_v$ is the sensible heat flux, and $LE_v$ is the latent heat flux from the canopy, all of which are given in $W \cdot m^{-2}$ (Oleson et al., 2013). Monin-Obukhov Similarity Theory (MOST) is used to determine resistances along the soil-plant-atmosphere continuum [Figure 3], which is then used to calculate $H_v$ and $LE_v$ (Zeng et al., 1998; Oleson et al., 2013). Using a big-leaf model, CLM represents both sunlit and shaded leaves (Dai et al., 2004).

The water balance equation tracks water flows through the system and connects to the energy balance via its dual controls on ET. The first of these controls, the influence of soil moisture on stomatal conductance, is not considered in this study. Soil moisture does not appear to limit stomatal conductance in the model; the predicted average value of transpiration wetness factor in CLM was typically around 95% in this study period and never fell below 50% for any 30-minute time period. Also, prior work has determined that ET at the present study site is not limited by soil water deficits during normal to above-normal rainfall years, such as the period from 2014 to 2016 (Andrews, 2016). On the other hand, leaf wetness can have an influence on this site. While its effect is considered to be small in some ecosystems (Burns et al., 2018), previous studies have shown that leaf wetness exerts significant influence on fluxes from rainforests in general (Loescher et al., 2005; Kume et al., 2011) and at this

site specifically (Aparecido et al., 2017; Moore et al., 2018). CLM reflects these mechanisms as well in the resistance network [Figure 3b], and the leaf wetness can prevent transpiration and contribute to canopy evaporation rates. Here, leaf wetness is determined by the interception rate of incoming precipitation (Deardorff, 1978; Dickinson et al., 1993; Lawrence and Chase, 2007). The amount of interception $q_{ic}$ is given in CLM4.5 as:

$$175 \quad q_{ic} = 0.25 \cdot q_{rain/snow} \cdot [1 - e^{-0.5(L+S)}] \tag{2}$$

and in CLM5 as:

$$q_{ic} = 1.00 \cdot q_{rain/snow} \cdot tanh(L+S) \tag{3}$$

where $q_{rain/snow}$ is the precipitation as liquid or snow, $L+S$ is leaf and stem area index, and 0.25 is a model coefficient. After determining intercepted rainfall, canopy water storage ($W_{can}$) is calculated through re-partitioning based on the condition of $0 \leqq W_{can} \leqq W_{max}$ ($mm$), where maximum canopy water storage ($W_{max}$) is $0.1(L+S)$ (Dickinson et al., 1993; Oleson et al., 2013). Finally, $f_{wet}$ is

$$f_{wet} = \left[ \frac{W_{can}}{W_{max}} \right]^{2/3} \tag{4}$$

Additionally, in CLM5, $f_{wet}$ cannot exceed a maximum value ($f_{wetmax}$) of 0.05. For instance, if $f_{wet}$ was 0.7, $f_{wet}$ would become 0.05. Finally, $f_{dry}$ is calculated as:

$$185 \quad f_{dry} = \frac{(1 - f_{wet}) \cdot L}{L + S} \tag{5}$$

In Eq. (4), the 2/3 exponent was assumed following the original literature (Deardorff, 1978), because the canopy water tends not to be evaporated when it is set to one and evaporates too fast when close to zero (Deardorff, 1978).

Additionally, CLM mainly uses the Farquhar model (Farquhar et al., 1980; Oleson et al., 2013) for photosynthesic rates. In our site, air temperature varies little throughout the year, and $CO_2$ concentration does not vary significantly. Consequently, light-limited photosynthesis can be considered as a dominant process. The light-limited model $w_j$ ($\mu mol \cdot m^{-2} s^{-1}$) in CLM is developed based on the Farquhar model (Oleson et al., 2013) and can be written as:

$$w_j = 0.25 J_x \mathbb{C}_i, \qquad \mathbb{C}_i = \frac{c_i - c_p}{c_i + 2c_p} \tag{6}$$

where $c_i$ is intracelular $CO_2$ concentration, $c_p$ is $CO_2$ compensation point, assuming 4 electrons per $CO_2$ molecule, $\mathbb{C}_i$ is a function of $c_i$ and $c_p$, and $J_x$ ($\mu mol \cdot m^{-2} s^{-1}$) is the electron transport rate which can be estimated through

$$195 \quad \Theta J_x^2 - (I_{PSII} + J_{max}) J_x + I_{PSII} J_{max} = 0 \tag{7}$$

where $\Theta$ is a curvature parameter ($\Theta = 0.7$ by default), $J_{max}$ ($\mu mol \cdot m^{-2} s^{-1}$) is maximum rate of electron transport. $I_{PSII}$ can be estimated as $I_{PSII} = 0.5\Phi \cdot I_{APAR}$, where $\Phi$ is quantum efficiency of photosystem II ($\Phi = 0.85$), 0.5 is for two photosystems for one electron, and $I_{APAR}$ is absorbed PAR ($\mu mol \cdot m^{-2} s^{-1}$).

To further explore these relationships, Eq. (6) and Eq. (7) are simplified and recalculated to make them comparable to apparent quantum yield ($\alpha$). This is because the light-limited model has a hyperbolic shape and the shape changes influenced by other environmental conditions. However, the apparent quantum yield is a slope parameter (or the initial slope of the light-limited model) between absorbed-PAR and photosynthetic rate, which is a well known and simple parameter with a long research history in the literature (Skillman, 2007; Evans, 2013). From Eq. (6), if ambient condition has $c_p \approx 40\mu mol \cdot mol^{-1}$, $c_a \approx 400$ and $c_i/c_a \approx 0.7$, it gives $c_i \approx 0.7 \cdot 400\mu mol \cdot mol^{-1}$ (Launiainen et al., 2011; Katul et al., 2010) and $\mathbb{C}_i$ becomes 0.667. If $c_i$ becomes higher as atmospheric $CO_2$ concentrations increase, it will approach 1. Through Eq. (6) and Eq. (7), the initial quantum yield of $CO_2$, also known as apparent quantum yield ($\alpha$), can be estimated via $\partial J_x / \partial I_{APAR} \times 0.667 \times 0.25$, which can be used with simple-version models such as $w_j = \alpha \cdot I_{APAR}$. It is worth noting that the differential has brought independence from $J_{max}$ at zero APAR, which is highly related to nitrogen and leaf temperature. The theoretical maximum for $\alpha$ should be $\approx 0.11$, $\alpha$ with saturated condition is approximately 0.075 (absence of photorespiration), and in normal atmosphere condition $\alpha$ is about 0.05 which is estimated if $\Phi \approx 0.6$ in Eq. (7) (Evans, 2013; Raj et al., 2015; Skillman, 2007). These light-limit models with different parameters are explored with observations in later section.

## 2.4 Simulation Setup and Comparison Method

CLM was tested in point-scale mode and the satellite phenology (SP) mode with default settings, with exceptions noted below. Extension modes, which consider additional processes such as dynamic global vegetation (DGVM), biogeochemical cycles (BGC), or carbon-nitrogen cycling (CN) were in general not considered since they do not affect our study interests here (e.g., tree growth and stand competition). Input parameters for the simulation were determined using the 'mksurfdata_map' utility provided in Community Earth System Model (CESM). The utility derives its values from satellite-based global datasets of phenology, soils, and topography, provided by University Corporation for Atmospheric Research (UCAR) (Oleson et al., 2013).

Based on multiple initial tests, we decided to use default parameters from the global surface data for our model, as varying them had no significant influence on model performance. Location specific default parameters from the global dataset include: leaf area index ($LAI$, $5m^2 \cdot m^{-2}$), stem area index ($SAI$, $0.8m^2 \cdot m^{-2}$), canopy height (34 m), sand clay loam soil (47% sand, 26% clay, 27% silt), organic matter density (33 $kg \cdot m^{-3}$), and color class (15). We need to note that default CLM cannot yet reflect Leaf Area Density (LAD) as in [Figure 2]. Changing any of these parameters from the global values to local values did not significantly affect the model's results in our tests. This is perhaps because our LAI value is high enough to be the dominant parameter, and the role of the soil is small. Moreover, the slope parameter exists in the model but it is not actually used in CLM's radiative transfer, canopy process, and turbulence sub-models. Additionally, most of the measured parameters at this site were not very different from the default values. Therefore, we decided to use the default setting except for some significant differences as outlined below. The tropical, broadleaf evergreen tree (BET) plant functional type (PFT) was used as the basis for representing the site's specific landcover. The location in question had a default value of 30% BET tropical, 30% of tropical broadleaf deciduous trees (BDT Tropical), and 25% for grass and crop which we altered to 100% BET for purposes

of this study. The atmospheric reference height was set to 44 m to reflect the location of the net radiation sensor on the Canopy Tower.

As an input, a meteorological forcing data set for CLM was created based on the measurements collected on site. These variables included half-hourly averages of wind speed ($m \cdot s^{-1}$), incoming solar radiation, relative humidity, air temperature, air pressure, precipitation, and $CO_2$ concentration. Comparison of the simulation was based on measurements taken at Canopy Tower; thus, Canopy Tower data was primarily used as forcing data when possible data was available. Average precipitation and air temperature data collected at 10-m height at Met Tower were also used for data gap-filling. In most cases, weather data obtained from the two towers were highly correlated, as the locations are less than 1 km away and only differ in their immediate surroundings (i.e., forest vs. clearing) and slope degree (i.e., ~45 degree slope vs. flat terrain).

Although flux methods cannot measure gross primary production (GPP) directly, it is an extremely important variable in the context of global carbon cycle modeling. In light of this, we estimated GPP based on net ecosystem exchange (NEE), net ecosystem production (NEP), and ecosystem respiration (ER), where NEE ≈ NEP and GPP = NEP – ER. With eddy-covariance data collected at the height of 33 m, NEP was estimated as $CO_2$ flux + $CO_2$ storage flux. Ecosystem respiration (ER) was estimated to be around 1.2 ($\mu mol \cdot m^{-2} s^{-1}$) based on the nighttime data found using the u* threshold method (Papale, 2006; Reichstein et al., 2005). This EC based data for $CO_2$ and $H_2O$ flux can be still questionable due to the instrument configuration. However, comparison of the EC data and sap-flow data (discussed below) showed acceptable similarity, and these data were accurate enough to give the information whether the model has a significant error.

For transpiration (TR), measured data and simulated transpiration rates are compared at daily timescales. For the comparison, it is necessary to estimate or measure each major flux (partitioned flux) within ET. In this site, up-scaled sap-flow data provides a transpiration rate (Aparecido et al., 2016), which in turns allows for water vapor flux partitioning. Although the sap-flow data at the site tends to lag temporally and nocturnal sap-flow occurs (shown later), it provides data to be used as a comparison at a daily scale against CLM estimates. As CLM cannot represent nighttime transpiration, the nighttime sap flow data, collected when the cosine zenith in CLM is less than zero, were eliminated from the comparison. However, taking into account that the nighttime sap-flow rate possibly occurs to recharge the sap water, an additional comparison was made without the elimination of nighttime value. This daily scale comparison is made by a one to one figure with R-squared value. Also, regression analysis provides additional information how much the model deviates from observation, as a slope of 1 and an intercept of 0 are expected from model/measurement comparisons. We note that the intercept is related to the daily average value, and it should be directly affected by the elimination of the nighttime transpiration and that a portion of this difference is related to the lag.

Unlike the radiative transfer models, CLM5 notably updated from CLM4.5 the physiological models for GPP and TR and their associated parameters. The Ball-Berry Model (BB) (Ball et al., 1987) was supplanted by a combination of the Medlyn model for the stomatal conductivity (Medlyn et al., 2011), a plant hydraulic stress model (Bonan et al., 2014), and the Leaf Use of Nitrogen for Assimilation (LUNA) routine (Ali, 2016). For the stomatal conductivity, regardless of the type of model (BB or Medlyn model), the slope parameter, which links stomatal conductivity and carbon fixation (i.e., photosynthesis), has been reduced by the model update. While the BB model still can be used for CLM5, its slope parameter has been changed from 9 to

7.3 for C3 plants. We have tested several options in CLM5 and determined that changing the stomatal conductivity model does not affect photosynthesis-related results (e.g., GPP) in our case.

To facilitate comparisons, CLM requires to assign height of each output variable. In this case, each reference height was determined based on given parameters in CLM: the displacement height was $d = 23.45m$, ground roughness height was $z_{0mg} = z_{0vg} = z_{0hg} = 0.01m$, and surface height was $z_0 = z_{0mv} = 2.625m$, so the canopy height became $d + z_0 = 26.075m$. For instance, canopy air temperature (Ta) in CLM was 2 m temperature in this comparison study, and it had $d + z_0 + 2 = 28.075m$. CLM uses $T_s$ term in Figure 3 which refers to canopy air temperature but the CLM module does not provide $T_s$ values as one of the output variables. This 2 m temperature (Ta), named as TSA in CLM ouputs, would be the closest available value from the canopy. Moreover, our profile data indicates that air temperature does not vary much in different height near the top canopy, and $28.075m$ is still within canopy. Our instrument heights did not exactly correspond to those heights from CLM, so the nearest one or two data points was used for the comparison rather than interpolating all data.

Additionally, CLM5 has a low default leaf wetness ratio; the maximum is 0.05 as in Eq. (4). For fair comparison, all leaf wetness values from CLM were normalized to a [0-1] scale based on the water amount on canopy using Eq. (4). Additionally, the question of whether or not to apply the power of 2/3 did not change our comparison results significantly.

Soil related data was spatially up-scaled and vertically interpolated to compare with the simulation. For the spatial up-scale, soil temperatures and soil heat fluxes were measured at five different places near the Canopy Tower, and the vertical profile data were also collected close to the base of the tower. For the vertical profile, CLM considers a larger number of soil layers. Therefore, the results of CLM were linearly interpolated, to compare with the measured data.

To initialize the simulations, CLM was first executed with a cold start (i.e., randomly produced initial values) and run for 100 years to get stable soil temperatures, cycling (through) the 6-year forcing data collected between the beginning of 2010 and the end of 2015. Once stable soil temperatures were obtained, CLM was rerun for two years (2014 – 2016) at a 30-minute time step. For some cases, linear regressions were performed to compare CLM outputs to field data. Goodness-of-fit of the regression analysis was determined based on coefficient of determination (R-squared) where appropriate. In this analysis, we focused on the following variables: net radiation, PAR, albedo, $CO_2$ flux, GPP, transpiration, latent heat flux, air temperature, leaf temperature, leaf wetness, and soil-related variables. We additionally tested how changes in levels of maximum leaf wetness ($f_{wetmax}$) and quantum efficiency of photosystem ($\Phi$) affected goodness-of-fit. Modifications of LAI, light extinction related coefficients, and canopy heights (34m~44m) were also tested. Unlike $f_{wetmax}$ and $\Phi$, however, they provided no significant difference or better results, so comparison and discussion of them are not made here.

## 3 Simulation Results and Comparison

### 3.1 Net Radiation and Albedo

A comparison of light-related variables indicated the simulated land surfaces received less energy than field measurements, but the difference was not significant. Simulated net radiation values were $20\ W \cdot m^{-2}$ less than the average measured values, although diurnal patterns closely matched ($R^2$=0.99). Net radiation in CLM was approximately 15 to 45 $W \cdot m^{-2}$ lower than

field measurements during the daytime and 10 to 15 $W \cdot m^{-2}$ lower during the nighttime [Figure 4a; Figure 4b]. Little difference ($< 5\,W \cdot m^{-2}$, $R^2$=0.99) was detected between CLM4.5 and CLM5. The simulated shortwave reflectance (albedo) in CLM was around 15% higher than the gauged albedo (+0.022 across all daytime data) [Figure 4c], which likely caused the differences in daytime net radiation.

Light data was clearly affected by the sloped terrain. Although the models were developed for all the global surface, sub-grid scale heterogeneity in land surface elevations has not yet been implemented in CLM4.5/5.0. Albedo from CLM tended to have a symmetric form, while the measured albedo had a skewed diurnal pattern [Figure 4c]. This skew caused a noticable discrepancy with the modeled values in the early morning which peaked during mid-afternoon (+0.0517 at 3PM; [Figure 4c]). The highest PAR intensity (or highest incoming solar radiation) occurred at 10 a.m. [Figure 4e], when the albedo difference between the observations and the simulation was smallest (+0.0214). In some parts, this may be caused by the too simple albedo models, which cannot properly respond to the intensity of solar radiation/angle. However, the skewed albedo seen on the measured data in [Figure 4c; Figure 4d] clearly indicates that CLM cannot represent the slope effect of the land surface. Such skewed diurnal variations were also observed in the PAR profiles [Figure 4e; Figure 4f]. The measured PAR values, generated by sensors somewhat shaded by the upper canopy, were diurnally skewed compared with shaded PAR from CLM. In contrast to the solar radiation above the canopy (i.e., the top of the tower, net radiation), the radiation profile started to become skewed right after infiltrating the top canopy layer. When revisiting the effect of canopy gaps created by the emergent tree, we observed that radiation values between the top of the canopy ($\approx 400\,W \cdot m^{-2}$ at 44 m from Net Radiation) and the next nearest heights ($\approx 110\,W \cdot m^{-2}$ at 32-38 m from PAR) were considerably different (about 70-80% reduction from the top) as mentioned before. The height of the primary canopy, consisting of the dominant trees, is about 38 m (Aparecido et al., 2016). Therefore, the shade effect (optical thickness) may be substantial, even though the emergent trees added minimal thickness. This feature can be important because the hill-slope surface is more sensitive to the sun angle , which affects the sunlit/shaded area ratio.

Simple manipulation was attempted by changing the solar angle to mimic the slope effect on albedo [Figure 4c; Figure 4d; Figure 4e]. The cosine zenith angle in the two-stream approximation was re-estimated by pushing back 30 degrees, to apply to the light extinction coefficient K in two-stream approximation. This simple modification reduced some the skewness of albedo [Figure 4d]. However, shaded PAR showed opposite behavior compared to the observation, mainly because sunlit area was increased.

## 3.2 CO$_2$ Flux (GPP)

All CLM versions (CLM4.5, CLM5, and CLM5BGC) overestimated GPP (6.7, 4.9, and 3.6 $\mu mol \cdot m^{-2} s^{-1}$) [Figure 5a; Figure 5b]. Results from the new version, CLM5, were generally more similar to the measured data than those in CLM4.5 [Figure 5a]. CLM5 yielded lower photosynthetic rates than CLM4.5, possibly due to the lower BB slope parameter, and also due to suppressed maximum rates of $V_{c,max25}$ and $J_{max25}$ by LUNA and BGC mode. Inactivating the plant hydraulic stress model in CLM5 increased the carbon-assimilation rate, while disabling the LUNA model decreases it in this site study. The prediction for the middle range of photosynthetic rate (5-15 $\mu mol \cdot m^{-2} s^{-1}$) did not improve much compared with CLM4.5.

One of the possible causes of the discrepancy between the estimated GPP and its observed values may be the model determining the response to light-limitations [Figure 5b]. Comparison between absorbed PAR (APAR) versus GPP shows that the initial slope of measured data is much lower than the simulated one [Figure 5c]. The APAR, including sunlit and shade leaf area, was estimated in CLM using measured incoming solar radiation above the canopy at 44 m. As previously explained [Figure 5d], an extensive literature study Skillman (2007) and Evans (2013) showed the theoretical maximum for $\alpha$ should be $\approx$0.11, that $\alpha$ under saturated conditions is approximately 0.075 (absence of photorespiration), and that in normal atmospheric conditions $\alpha$ is about 0.05, which is estimated when considering $\Phi \approx$0.6 in Eq. (7) (Evans, 2013; Raj et al., 2015; Skillman, 2007). From our observations, the fitted value for $\alpha$ was 0.021 ($\Phi \approx$0.25). This low value may have been caused by other factors such as physiological stress or a scale problem. The fitted value was estimated from eddy-covariance measurement rather than at the leaf-scale. By default, the $\alpha$ is around 0.07 in CLM4.5 and CLM5, with $\mathbb{C}_i$=0.667, which is higher than 0.05 as usually reported (Skillman, 2007; Ehleringer and Pearcy, 1983; Ehleringer and Björkman, 1977). Of course, this method itself has a possible error caused by the issues inherent in the eddy-covariance measurements, as well as the estimation of APAR in CLM, which contains only sunlit and shade leaf area, making it too simplistic. For this study, $\Phi$ was modified to produce an appropriate value for $\alpha$, but the issue should be revisited in future studies.

Test simulations with CLM4.5 and CLM5 were conducted using $\Phi$=0.25 and $\Theta$=0.7. When $\Phi$ was updated, both CLM4.5 and CLM5 performed better than before [Figure 6;Figure 5a]. This change resulted in more stable predictions, as judged by the middle range of GPP (5-15 $\mu mol \cdot m^{-2}s^{-1}$). Maximum GPP was reduced as expected [Figure 5d], and it was possible to further fix such over reduction by updating $\Theta$ (curve shape), as shown on [Figure 5d]. The change of the slope can alter maximum assimilation rate, and the alteration can be counterbalanced if $\Theta$ is modified. In the simulated diurnal variation plot, the trend was slightly shifted in the afternoon, also probably due to the effects of the topographical slope [Figure 6b]. Time-dependent classification (i.e., regression lines with intercept forced through zero [Figure 6b]) and the fitted slopes indicated that geographical features have an influence on photosynthetic activity, which is mostly caused by the radiative transfer models, like albedo. However, the model failed to accurately represent such features, since $CO_2$ flux in CLM was lower in the morning and higher in the afternoon.

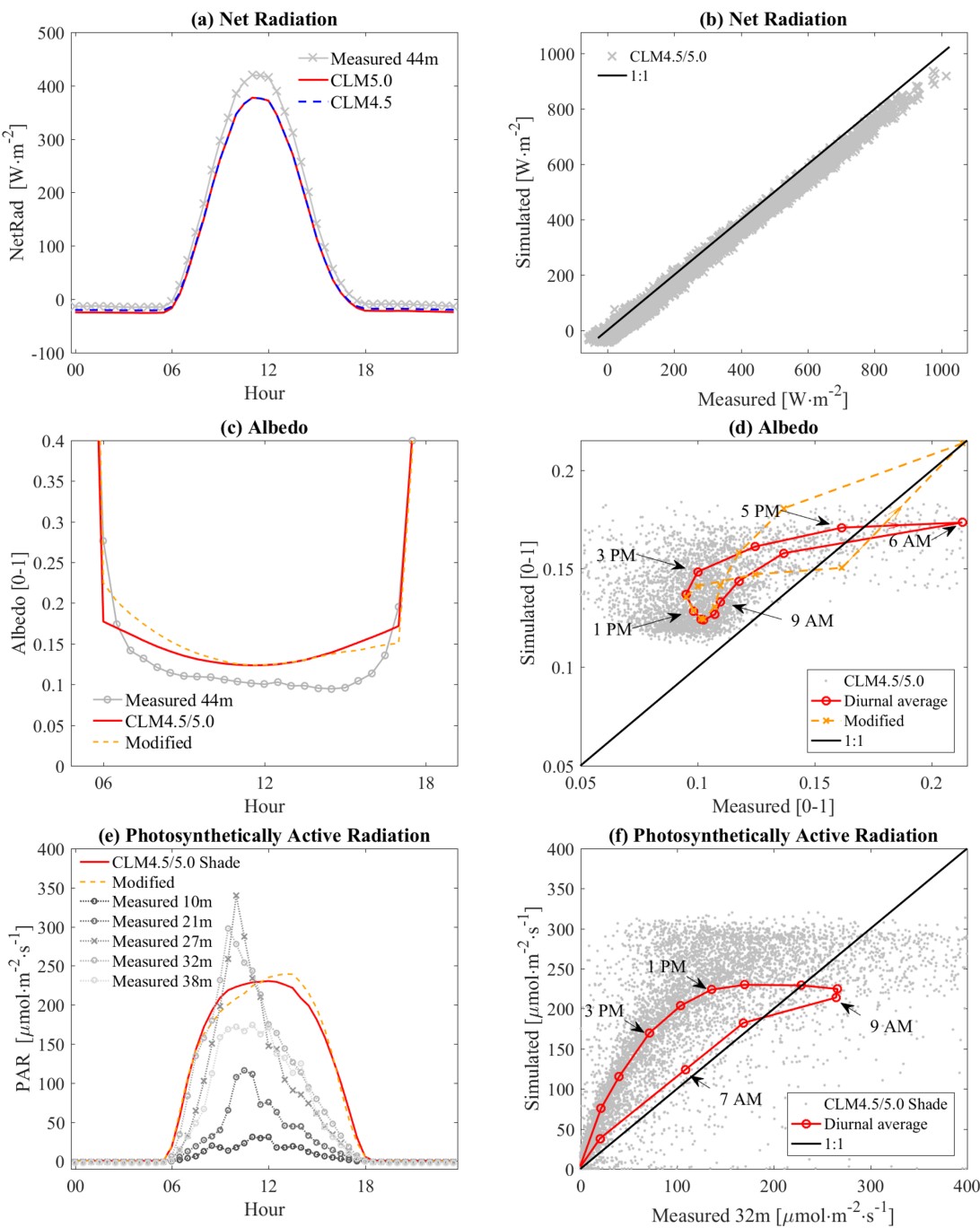

**Figure 4.** (a) and (b) - comparison of net radiation between CLM and measurement on Canopy Tower at 44m. (c) and (d) - albedo at 44m. (e) and (f) - PAR comparision for shaded canopies. All left plots (a, c, and e) are ensemble diurnal variation and the right plots (b, d, and f) are one to one comparison plots between CLM and measured data. Hysteresis depicted on (d) and (f) is based on hourly ensemble average values for daytime. The 'Modified' is an attempt to mimic the slope effect by the simple update of the two-stream approximation.

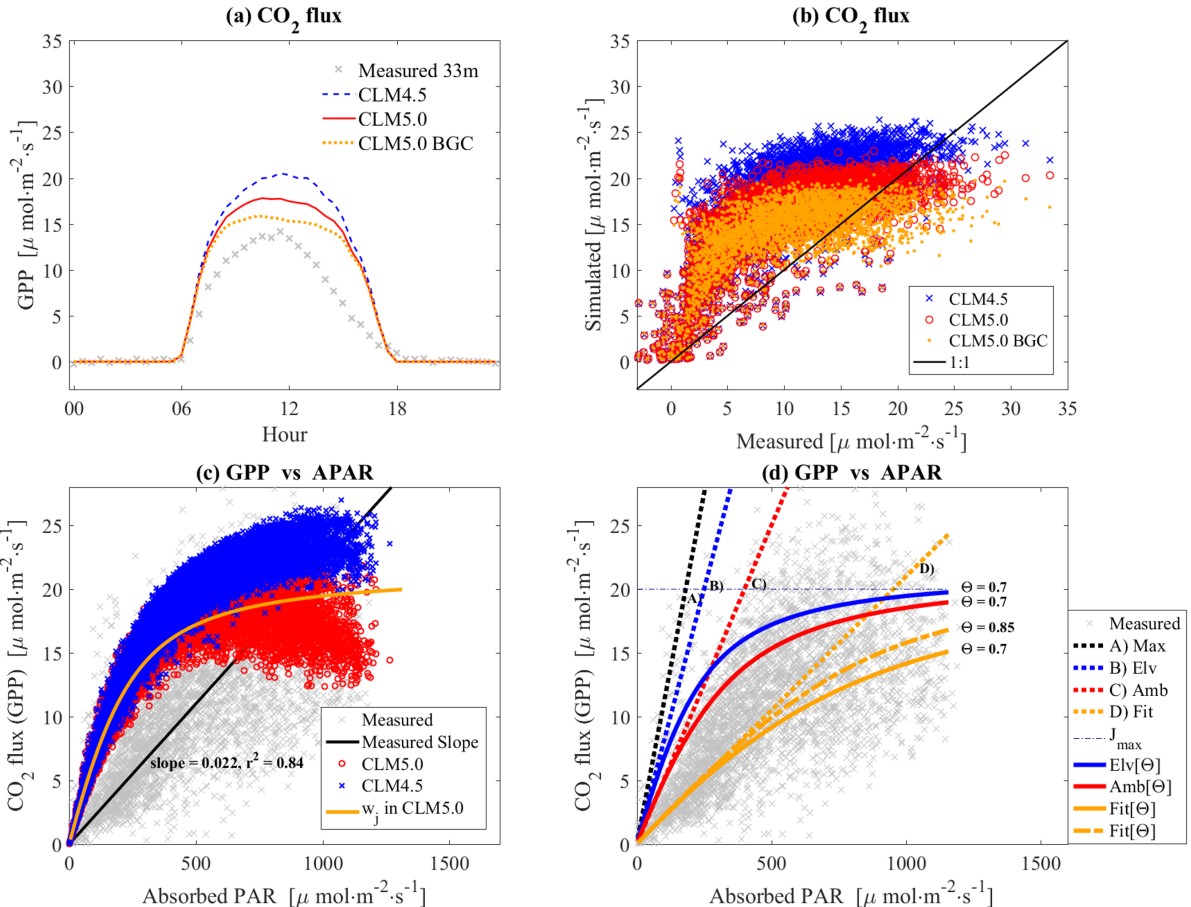

**Figure 5.** (a) The ensemble diurnal variation of $CO_2$ flux: differences between Eddy Covariance (Canopy Tower 33m) and CLM in daytime is 6.7, 4.9, and 3.6 $\mu mol \cdot m^{-2} s^{-1}$ for CLM4.5, CLM5, and CLM5BGC, respectively; (b) One-to-one plot in reference to data shown in figure (a); (c) APAR vs GPP, and $w_j$ is simulated with default parameters and $\mathbb{C}_i$=0.667; (d) The light-limit model tested with different parameters. 'Max' is the theoretical maximum, 'Elv' is saturated/elevated condition, 'Amb' is normal atmosphere condition, and 'Fit' is a fitted value from our observation. In the legend, '[Θ]'means the usage of hyperbolic function Eq. (7), like 'Elv[Θ]'. Without '[Θ]', only the slope parameter is active as 'B) Elv'. These parameters are described in full in Sections 2.3 and 3.2.

## 3.3 H$_2$O Flux

The effect of the change of $f_{wetmax}$ were detected in the model's results for vapor fluxes [Figure 7]. Again, CLM5 has a low leaf wetness coefficient (i.e., maximum rate is 0.05 as Eq. (4), which reduced canopy evaporation and elevate transpiration rate). In this simulation, $f_{wetmax}$ was considered as 1 for CLM5 (hereafter referred to as CLM5 fmx=1), and we used this when we wanted to make a fairer comparison with CLM4.5

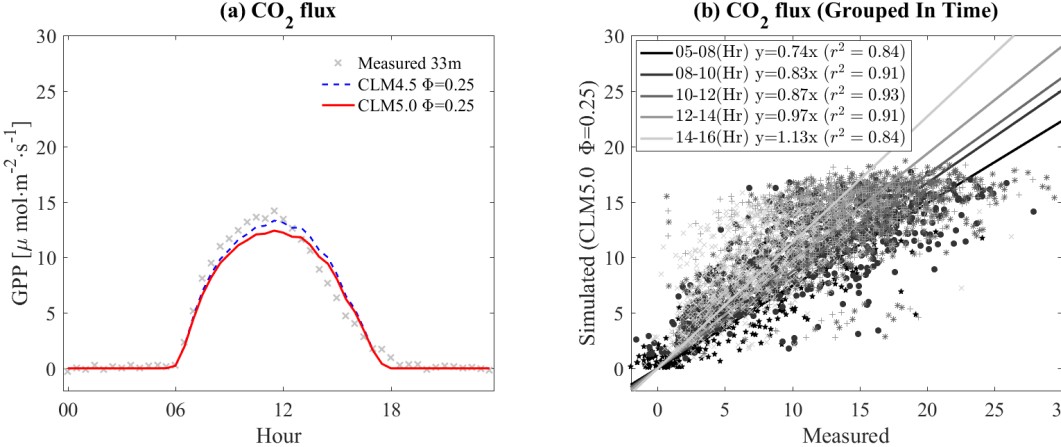

**Figure 6.** Test simulation using $\Phi=0.25$ and $\Theta=0.7$ as in [Figure 5d]. By the modification of $\Phi$, maximum GPP has been reduced. It is possible to improve this model by updating $\Theta$. $r^2$ is a R-squared value without an intercept. The units for both (a) and (b) are the same as $\mu mol \cdot m^{-2} s^{-1}$.

Similarly to $CO_2$ flux, total $H_2O$ fluxes of CLM5 were overestimated ($2.1 \times 10^{-5}$ $mm \cdot s^{-1}$ higher in daytime than eddy-covariance values). Flux rates in the CLM5 fmx=1 simulations were reduced in comparison to those predicted by CLM4.5 [Figure 7a; Figure 7b]. The notable decrease (CLM4.5 & CLM5 with $\Phi=0.25$) was due to the change of the quantum yield $\alpha$ parameter needed for GPP predictions [Figure 6]. Transpiration rates (TR) also showed similar trends, and this indicates that TR is an important process in this site.

At the daily time scale, CLM4.5 produced the highest estimates for both ET and TR in comparison to the other versions [Figure 7e; Figure 7f]. CLM5 yielded a notable reduction of ET and TR due to the newly implemented leaf wetness parameter $f_{wetmax}$. We can visually identify that applying a quantum efficiency of $\Phi=0.25$ made fitted lines closer to the 1:1 line for both ET and TR [Figure 7e; Figure 7f]. However, we cannot conclude that it was improved, since although the low $\Phi$ changed the slope and intercept values it did not change the R-squared values significantly. This change might also be influenced by other components such as leaf wetness. Here, correlations of TR were slightly increased, by around 0.01 ($R^2_{CLM4.5} = 0.67$, $R^2_{CLM5}$ = 0.68), when considering $\Phi = 0.25$. On the other side, the correlations of ET were decreased by around 0.1 ($R^2_{CLM4.5} = 0.42$, $R^2_{CLM5} = 0.44$) [Table 1]. When assuming a lower quantum efficiency, the change in TR makes the fitted slope for ET decrease [Figure 7e], possibly since transpiration is a more influential component than evaporation in this site. Thus, TR drove ET rates when there was higher energy exchange condition (i.e., warm, sunnier and drier time). On the other hand, these results also highlighted the importance of other sub-models such as canopy evaporation.

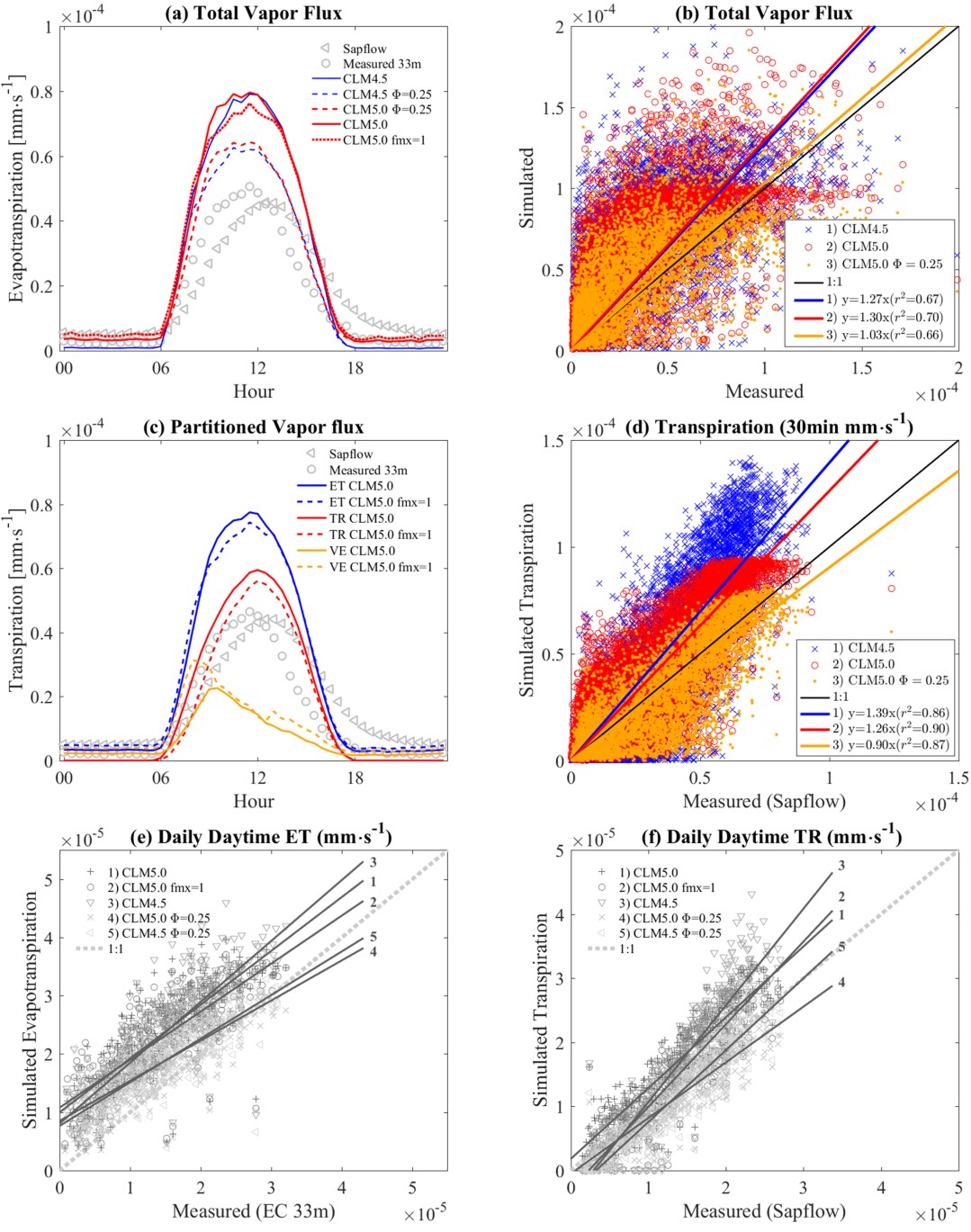

**Figure 7.** (a) The ensemble diurnal variation of total H$_2$O flux, where "Measured 33m" is measured by Eddy Covariance (at 33m), "Sapflow" is transpiration measured through sapflow, all "CLM" are about evapotranspiration (ET), 'fmx=1' sign represents fwetmax=1, and 'Φ=0.25' means 0.25 applied to Φ in Eq. (7); (c) Partitioned H$_2$O flux, where ET, TR, and VE are evapotranspiration, transpiration, and canopy evaporation from CLM; (b, d) The one to one plots of (a) and (c); (e) and (f) Daily ET and TR (except nighttime) against eddy-covariance and sapflow data.

**Table 1.** Fitting parameters and regression coefficients for sap flow and eddy-covariance measurements versus simulation by CLM in daily scale for [Figure 7e] and [Figure 7f]. The nighttime data are excluded (set to zero) for both and values in parentheses are with nighttime data. The unit for the intercept is $10^{-6} \cdot mm \cdot s^{-1}$.

| Figure | Line | Model | Data | Slope | Intercept | $R^2$ |
|--------|------|-------|------|-------|-----------|-------|
| [Figure 7e] | 1 | CLM5 | EC 33m (ET) | 0.92 | 9.98 | 0.51 |
| | 2 | CLM5 fmx=1 | | 0.82 | 10.79 | 0.51 |
| | 3 | CLM4.5 | | 1.04 | 8.06 | 0.55 |
| | 4 | CLM5 Φ=0.25 | | 0.69 | 8.51 | 0.42 |
| | 5 | CLM4.5 Φ=0.25 | | 0.75 | 7.65 | 0.44 |
| [Figure 7f] | 1 | CLM5 | Sapflow (TR) | 1.11 (1.01) | +1.79 (-0.22) | 0.80 (0.66) |
| | 2 | CLM5 fmx=1 | | 1.29 (1.17) | -3.00 (-5.32) | 0.81 (0.67) |
| | 3 | CLM4.5 | | 1.51 (1.37) | -4.43 (-7.29) | 0.81 (0.67) |
| | 4 | CLM5 Φ=0.25 | | 0.87 (0.79) | -0.51 (-2.14) | 0.80 (0.67) |
| | 5 | CLM4.5 Φ=0.25 | | 1.12 (1.03) | -3.67 (-5.84) | 0.80 (0.68) |

## 3.4 Leaf Wetness

The leaf wetness fraction predicted by CLM was compared to observations made using capacitance sensors [Figure 8]. In the analysis, the ensemble diurnal variations of leaf wetness were plotted, where '38m', '11m', and '3m' are measurement heights and the others are leaf wetness from CLM5 ($f_{wetmax}$=0.05), CLM5 fmx=1 ($f_{wetmax}$=1), and CLM4.5 ($f_{wetmax}$=1) [Figure 8b]. The predicted leaf wetness was not in agreement with the diurnal leaf wetness variation measured at this site [Figure 8b]. In particular, the night-time fraction of leaf wetness was significantly higher when compared with gauged data. The biggest problem detected in this study was that intercepted canopy water was rarely evaporated in the model. The canopy water tended to accumulate, especially due to frequent nighttime rainfall that started in the late afternoon or high daytime humidity which caused condensation. Daytime leaf wetness seems to be reasonably simulated [Figure 8b]. However, no trend could be identified in the comparison between simulated and measured data (not displayed here), which indicated that the formula cannot adequately represent actual behaviors of wet fraction in both CLM5 and CLM4.5.

Intercepted precipitation was usually too high in CLM compared to observed leaf wetness [Figure 8c; Figure 8d]. The values in [Figure 8c] and [Figure 8d] show the increasing rate of leaf wetness due to precipitation, with the large and thick markers indicating their averages. The collected data was conditioned upon the absence of a rainfall event at least 2 hours prior to and an initial leaf wetness lower than 0.2. [Figure 8c] shows 0.5-hour rainfall events (one consecutive event in 30-min scale) and [Figure 8d] is for 2 hours rainfall events (four consecutive events). This increment was directly related to canopy interception: the usual increment for 2-hour (and 30-min) rain was 0.71(0.33) at a 38-m height , 0.48(0.28) at a 3-m height , around 0.88(0.73) in CLM5, 0.97(0.77) in CLM5 fmx=1, and 0.94 (0.46) in CLM4.5. The modified interception model (CLM5 fmx=1) from Eq. (3) resulted in higher interception rate than CLM4.5 fmx=1 [Eq. (2)]. The interception rate also seemed higher with CLM5 fmx=1 than with original CLM5 as in [Figure 8c] because CLM5 fmx=1 had a higher canopy evaporation rate. This effect

resulted in the acceleration of canopy evaporation while allowing interception to play a larger role in the canopy water balance. In the one-to-one comparison, the increase of leaf wetness in CLM was usually higher than in measured data. Consequently, the wet canopy fraction at the beginning of the drying process was usually higher in CLM than in the measurements: 0.63 at the 38 m observation, 0.47 at the 3 m observation, 0.96 in CLM5, 0.9 in CLM5 fmx=1, and 0.78 in CLM4.5 (see y-axis data at 0 x-axis in [Figure 7e]).

Another analysis showed that leaf wetness behavior is highly sensitive to incoming solar radiation. The water state on a leaf in [Figure 8e; Figure 8f] was tracked over consecutive no-rain events for 3 hours just right after last rain events in the daytime between 10:00 AM and 14:00. [Figure 8e] shows events with low solar radiation (0 - 300 $W \cdot m^{-2}$) and [Figure 8f] shows events when solar radiation was higher than 300 $W \cdot m^{-2}$. Although it was difficult to gather data for these serial drying events (each plot uses at least 12 groups of 6 consecutive half-hourly time with no rain), the result clearly indicated that leaf wetness is strongly influenced by an increase of incoming solar radiation when $f_{wetmax}$=1 (CLM5 fmx=1 and CLM4.5). In the case of $f_{wetmax}$=0.05 (CLM5), the drying rate is reasonable at low solar radiation, but it is higher than values observed during high incoming solar radiation. The measured data in the analysis showed relatively smaller values of leaf wetness at lower levels of the canopy. This indicated that rainfall does not frequently reach the lower canopy, and thus interception rates are low there. This finding would suggest that lower $f_{wetmax}$ values are reasonable.

## 3.5 Temperatures and Soil Flux

The simulated canopy air temperature in both CLM4.5 and CLM5 was overestimated during daytime (+0.8 and +1 $C^{\circ}$, respectively) and underestimated during the nighttime (-1.9 and -1.1 $C^{\circ}$, respectively) [Figure 8]. In other words, the simulated temperature may be overly sensitive to incoming solar radiation, like leaf wetness, which was overestimated during the day and underestimated at night. The updated MOST scheme improved nighttime air temperatures in CLM5 (Burns et al., 2018), but they were still underestimated. As reported in the previous section, water remaining on the canopy during nighttime tended to be inefficiently evaporated [Figure 8b], which was also possibly related to low canopy temperature in CLM. At lower canopy levels, the ground air temperature at the surface was overestimated during daytime, and it was even higher than air temperature at heights of 1 m - 5 m [Figure 9b].

The ground surface tended to have high energy exchange during daytime similar to the canopy processes. Considering the soil temperatures [Figure 9] and the soil heat fluxes [Figure 10], we found they were overestimated during daytime and underestimated in the nighttime. Soil temperature and heat flux in CLM was highly variable. Soil evaporation rates in both CLM4.5 and CLM5 were also overestimated compared with estimated data from soil respiration chamber measurements (LI8100) [Figure 10]. For daytime soil evaporation, the average difference from the observation was $5 \times 10^{-7}$ $mm \cdot s^{-1}$ with CLM4.5 and $15 \times 10^{-7}$ $mm \cdot s^{-1}$ with CLM5. The measured field value was around $1 \times 10^{-7}$ $mm \cdot s^{-1}$. The simulated soil moistures also had high variability with low mean water contents (around 0.2 $m^3 \cdot m^{-3}$) compared with gauged values (0.3-0.4 $m^3 \cdot m^{-3}$).

The overestimation of vegetation temperature (Tv) in both CLM4.5 and CLM5 also appeared in the daytime simulation ($\approx$ + 1.0~2.4 $C^{\circ}$) [Figure 11a; Figure 11b]. Another model test was also made using global forcing datasets (Qian et al., 2006) to corroborate our simulations, and the result showed very similar behavior ($\approx$ + 5 $C^{\circ}$, not depicted here). The high Tv and Ta

from CLM simulations resulted in lower relative humidity than gauged-based canopy air humidity. We note that the sunlit/shade scheme in CLM does not consider two different vegetation temperatures, so it only takes a single variable Tv to represent the entire canopy. Canopy temperature (Tv) in CLM should be an average of sunlit and shaded leaf temperature but the simulated results were far from our expectation [Figure 11a]. A comparison plot also showed significant error [Figure 11b]. The additional comparisons indicate that Tv on sunlit leaves normally followed the canopy air temperature (leaf thermoregulation) but CLM did not reproduce such behavior [Figure 11c; Figure 11d].

## 4 Discussion and Conclusions

In this study, two versions of the Community Land Model (CLM4.5 and CLM5), running primarily in the satellite phenology (SP) mode, were tested against measured data from a mountainous tropical rainforest in Costa Rica. Net radiation was under-predicted by an average of -20 $W{\cdot}m^{-2}$ [Figure 4a; Figure 4b] in both CLM4.5 and CLM5. The discrepancy was attributed to CLM's over-prediction of surface albedo, which was, on average, 0.022 lower in the measurements [Figure 4c; Figure 4d].

The effects of topographic slope clearly appeared in the diurnal plots for albedo/PAR [Figure 4] and for $CO_2$ flux [Figure 6]. With respect to albedo, the hillslope shading effect magnified these discrepancies, with afternoon values having larger differences as the sun moved behind the north-south trending mountain [Figure 4]. The level of discrepancy varied according to the diurnal cycle of the intensity of incoming solar radiation and the solar angle [Figure 4c; Figure 4d]. PAR profiles also showed radiation levels within the canopy had a skewed, or hysteretic, cycle [Figure 4e; Figure 4f], which was not captured by CLM. These results indicated that canopy radiative transfer, including the surface albedo and sunlit/shade separation, may need to be better represented in advanced land surface models, in order to simulate a more realistic response to solar radiation or topographical slope. A simple modification of albedo was attempted, but it resulted in an error on other variables such as PAR. The update may require more complicated manipulation to satisfactorily match variables in addition to albedo (e.g., PAR). This finding suggests that a multiple layer scheme is necessary to properly represent light penetration. More importantly, aerodynamic resistance models, such as MOST, are also currently incapable of representing a sloped terrain. If the effects of both can be implemented in CLM, predictions can be highly improved for mountainous regions, especially if they can be considered at a fine grid scale.

The study found that slope affected various data and outputs to an important degree, and suggested that additional obser-vations are necessary to examine and capture such features. Several past studies to compare and improve CLM have taken a similar approach. However, they focused on specific sub-model performance (Burns et al., 2018; Swenson and Lawrence, 2014; Bonan et al., 2011), rather than studying the effects of spatial complexity. For albedo, the slope effects were minor in this study; skewness in the diurnal average curve was relatively small, and it is difficult to identify the difference between measured and modeled net radiation curves. On the other hand, the skewness for PAR is significant, and this was obviously related to the different response of GPP through time [Figure 6]. Such influence might not be noticeable if the GPP compar-ison were not classified by time, because the error appears similar to white noise. If this effect is captured, the prediction of physiological variables (e.g., GPP and TR) can be improved. We anticipate the same effect would be present in a wider range

of forests. Also, recent land surface models are becoming more elaborated by reflecting vertical (e.g., multi-layered canopies (Bonan et al., 2018; Ryder et al., 2016)) and horizontal (e.g., vegetation demographics (Fisher et al., 2018) heterogeneity. The performance of these advanced models would be affected by topographical characteristics. Hence, further investigation should focus on both improved model parameterization for hillslopes and additional data from mountainous forests.

The simulated photosynthesis rates tended to be higher than those observed; these result have also been reported in similar
studies of montane rainforests (Fan et al., 2019b; Muñoz-Villers et al., 2012). Such errors could possibly be alleviated by updating parameters associated with light-limitation effects. For carbon flux (GPP) and transpiration (TR), the over-estimation in CLM4.5 has been reduced in CLM5 [Figure 5b; Figure 7b; Figure 7d]. However, CLM5 and CLM-BGC seem to reduce the maximum assimilation rate limit by lowering the BB slope and other photosynthesis-limiting parameters (i.e., $V_{c,max25}$ or $J_{max25}$). The curved-shape error in GPP, at the middle range of photosynthesis rates still exists compared with CLM4.5 [Figure
5b]. At this point, we suggest that the light-response photosynthesis model could be the cause. We briefly addressed the electron transport model (Eq. (6) and Eq. (7)), and tested it by changing quantum efficiency and curvature parameters [Figure 5c; Figure 5c]. The analysis of GPP and transpiration values showed that changing the fitted quantum efficiency parameter resulted in better agreement with the observations and the effect of topographical slope appeared more clearly [Figure 6; Figure 7f]. The analysis contains possible errors caused by the simplified model for APAR and measurement error for GPP.

Partitioning the water flux is a critical task and this needs more investigation of each sub-model. Errors in vapor flux were particularly difficult to diagnose since the discrepancy can be caused by the failure of any of the embedded sub-models, although transpiration is the largest driver of the overall pattern of total vapor flux (ET) [Figure 7c]. Evapotranspiration (ET) consists of three major components: soil evaporation, canopy evaporation, and transpiration. Therefore, an error in any one of the sub-models can make the entire water flux (ET) inaccurate. We can also recognize that the comparison of total vapor flux
[Figure 7b] has much more uncertainty than $CO_2$ flux [Figure 5b].

Among the sub-models, canopy evaporation was key to proper partitioning for this site, and the process relies on both the rainfall interception sub-model and the leaf wetness sub-model. Both ET and TR were affected by the canopy evaporation [Figure 7a; Figure 7c], because leaf wetness suppressed transpiration and enhanced canopy evaporation in CLM [Figure 3b; Figure 8a]. However, the leaf wetness variable in CLM caused a high degree of uncertainty in a number of analyses, including
490 ensemble diurnal variation [Figure 8b] and interception rate [Figure 8c; Figure 8d], possibly due to too simple throughfall processes as reported in a previous study (Fan et al., 2019b). In Eq (3), when the leaf-stem area index is high ($L + S > 2$) the interception rate approaches 100% in CLM5. This value is questionable in our view because of the canopy in this site. The observed tree having high LAI (far higher than 2 ($m^2 \cdot m^{-2}$)) does not cover 100% of the sky ($\approx tanh(2)$). On the other hand, the value of 0.25 in Eq (2) for CLM4.5 seems too low. Leaf wetness related parameters are also optimized for large-scale
forcing (e.g., 6 hourly data). The improperly modeled canopy water levels and the wetted fraction resulted in errors in canopy evaporation which overreacted to the intensity of solar radiation or net radiation [Figure 8e; Figure 8f]. We observed some improvement in CLM5 by low maximum wetness $f_{wetmax}$ but the simulated leaf wetness was still sensitive to the incoming solar energy. We have tested more complicated interception models (e.g., Aston (1979)), but they resulted in only a small difference in the leaf wetness. Such water-related processed can have vertical/spatial variation due to the structure and the

shape of canopy and to the sloped topography. Our observations also showed variations in behavior based on height within the canopy, and such changes imply that more layers are necessary for accurate predictions of canopy water storage.

The new maximum leaf wetness applied in CLM5 may need to vary more by vegetation and leaf morphology, as highlighted in a previous study (Fan et al., 2019b). Changing $f_{wetmax}$ had a significant impact on latent heat fluxes [Figure 7a; Figure 7c; Figure 7e; Figure 7f], contrary to the results noted by Burns et al. (2018). This effect could be attributed to much more frequent rainfall at our site. Also, a low $f_{wetmax}$ is more reasonable for needle leaf species than it is for those with large, broad leaves. Leaf surfaces within the canopy cannot be easily fully-wetted even in this tropical forest. However, simply applying $f_{wetmax} = 0.05$ for all sites cannot be realistic. The role of leaf wet faction is not negligible in CLM, and the photosynthesis is still sensitive to leaf wetness ($f_{wet} \leqq 0.4$). At low relative humidity, the role becomes stronger [Figure 8a]. In our site, different leaf wetness behaviors have been observed between upper and low layer of the canopy (Andrews, 2016), which may also be an important characteristic for tracking leaf wetness, canopy evaporation, and ET.

From the similarity of two observations (EC vs. TR), we suspect the influence of a near emergent tree on the EC measurements, which is possibly diagnosed by the advanced model (e.g., profiled simulation). Such interference by the up-slope tree can occur anywhere in a sloped area and the CLM insufficiently represents spatial variability. Also, the TR was estimated using more than 40 trees with a 2200 $m^2$ plot. However, the footprint of the EC measurement does not necessarily match the area of the tree plot. In this case, incorporating a demographic model and a multi-layer model could provide a more appropriate basis of comparison for the TR and EC data. These additions might resolve the spatial scale issue and provide a method to handle some heterogeneity in the canopy (e.g., the emergent tree) beyond the traditional land surface model.

Temperature-related variables were also problematic in CLM [Figure 9; Figure 10; Figure 11]. This issue may be caused by errors in energy partitioning, modeling aerodynamic resistance, and physiological regulation. Daytime versus nighttime changes in canopy air temperature and leaf temperature in CLM were found excessive. . During the daytime, when peak temperatures occurred, the modeled relative humidity was low in the canopy airspace. These two variables could affect other physiological simulation results, such as transpiration rates. These errors may be caused by turbulence. In the Burns et al. (2018) study, changing the MOST parameters partially corrected underestimates of nighttime air temperature in CLM5. Other researchers have attributed these issues to incorrect parameterization of roughness length for heat and have made a number of advances toward reducing these temperature errors (Yang et al., 2002; Wang et al., 2014; Chen et al., 2010; Zheng et al., 2012; Zeng et al., 2012). However, we noted that our case is different since most studies discussed low diurnal variations and underestimations. The source of error was similar to, and perhaps intertwined with, the issues found with leaf wetness or other physiological regulation. The one-to-one comparisons between the canopy air temperature and the leaf surface temperature [Figure 11c; Figure 11d] indicated that Tv on sunlit leaves normally followed the canopy air temperature (i.e., leaf thermoregulation), as described in other literature (Michaletz et al., 2016). However, CLM does not consider such leaf thermoregulation processes. Consequently, soil temperature and all soil fluxes in CLM also had a higher degree of daily fluctuation than their measured counterparts [Figure 10]. These variables (soil moisture, soil temperature, and soil heat flux) are highly related each other, and they are also linked with the canopy condition (canopy air temperature, relative humidity, and aerodynamic resistance). Therefore, it was difficult to precisely diagnose the cause of such high variation.

Adjustments in light-related parameters (e.g., LAI, leaf angle, and optical depth) did not noticeably improve model results. The ratio of the absorbed energy on the soil surface to the total incoming solar radiation in CLM was 0.03, but our PAR profile data [Figure 4e] indicated the ratio should be lower, around 0.01. The average incoming solar radiation in the daytime was around 300 $W \cdot m^{-2}$. Estimated absorbed energy on ground and vegetation in CLM and the received energy at 10 m PAR sensor (unit was converted) were 9.4, 252.5, and 3.1 $W \cdot m^{-2}$. Even though the modeled ground surface tended to receive excess solar energy, changing this value did not seem to result in significant improvement in any simulated variables, because it was a relatively low portion of the energy budget. Likewise, increasing LAI to 7.7, based on nearby site measurements (Teale et al., 2014), only slightly alleviated issues associated soil temperatures and made no difference in canopy temperatures. We have also tested different leaf angles, which are directly related to the optical depth ($K$), but there was no significant difference; a change in leaf angle from $\chi_L$=0.1 to $\chi_L$=0.4 resulted in a 0.3 $C^\circ$ decrease in ground surface temperature. These supplementary tests indicated that the reduction of absorbed solar radiation on the ground and the some changes of parameters for soil albedo did not significantly alter canopy temperatures. The problem may more likely be caused by errors in the aerodynamic resistance above the canopy or too simplistic canopy structures, as has been reported in other studies (Wang et al., 2014; Chen et al., 2010; Zheng et al., 2012; Zeng et al., 2012).

A more complex big-leaf (two-layer) scheme may be necessary to improve the model. We find that both CLM5 and CLM4.5 used a two big-leaf scheme to describe the differences between sunlit and shade areas in the canopy. However, while this module partitions incoming solar radiation, it does not calculate the resulting differences in leaf temperature. Partitioning leaf temperature into sunlit and shaded values may be a promising adjustment due to the fact that the two have somewhat different behaviors. This effect was evident in measured versus modeled vegetation temperatures [Figure 11a]; the fraction of sunlit LAI for these plots was about 26% in CLM). The fraction of leaf wetness also represents the entire canopy area in CLM, which seems too simplistic. Maybe, the sunlit area should intercept the precipitation first, and dry out faster than the shaded area. On the other hand, this two-layer scheme still involves up-scaling issues to capture in-canopy variability such as the vertical segmentation of the light, physiological parameters, and the energy exchange (Bonan et al., 2011; Wang and Leuning, 1998; De Pury and Farquhar, 1999).

Beyond this two-layer structure, full profiled models, including momentum and mass conservation scheme with storage flux, would be much more promising. It was not difficult to identify the vertical variability of micrometeorological variables through observations (Andrews, 2016). For instance, the higher locations in the canopy tended to be more easily wetted/dried than the lower locations; the more exposed canopy area (higher location) was normally wetter than shaded canopy area [Figure 8e; Figure 8f]. Schemes with many layers can better simulate the full range of temperature, leaf wetness, and net radiation, which can naturally give a more realistic function of $f_{wetmax}$, temperature, interception, and physiological behavior compared to such single or two-layer scheme. The structure update such applying a LAD profile ($LAI_z$) as in [Figure 2] should be done first before re-parameterizing other sub-models. Such a multi-layer scheme would be a bridge between leaf scale parameters and canopy scale simulation. Additionally, adding storage flux can be influential in the tall, dense canopies of rainforests (Heidkamp et al., 2018), since the storage flux was not implemented in CLM. The heat storage can be related to air under

the canopy, but the role of vegetation biomass is significant; adding heat storage of vegetation biomass reduced the diurnal temperature range (Swenson et al., 2019; Meier et al., 2019).

    In conclusion, we have tested CLM's predictions of land-atmosphere processes in a mountainous tropical rainforest. This study determined the degree to which global-scale parameterizations work at this unique site. Very few case studies like this are currently available, and these results have provided some unique insights. We found that CLM5 has some advantages over
CLM4.5 under wet and steep conditions. However, CLM5 does not yet sufficiently resolve a number of critical problems, such as in the partitioning the energy. Model updates to the representation of in-canopy processes and features - namely photosynthesis, turbulence transport , canopy structure - are still needed to capture temperature variations and physiological activity. More importantly, further investigation into including terrain slope effects into the models is required.

    Additionally, we found that canopy temperatures and leaf temperatures were over-sensitive to incoming solar radiation.
These errors caused a number of cascading issues: low relative humidity near the canopy surface, subsequently affecting tree physiological processes, and excessive heating of the soil surface, leading to unrealistically high average and day-to-night differences in soil temperatures and soil heat fluxes. The formulation describing leaf wetness processes is too simplified, which caused model failure for the frequently rainy areas. Transpiration rate, which was the largest part of latent heat flux at the site, as well as carbon uptake through photosynthetic activity, were also over-estimated in CLM. In the photosynthesis model,
quantum efficiency also needs to be re-parameterized. Other attempts such as slope effect to a radiative transfer scheme and more complex interception model did not give significant improvement. Ultimately, however, it may be necessary to apply a complete big leaf scheme (two-layer scheme) or multi-layer scheme to better depict the multi-faceted interactions between leaf wetness, temperature, and shading to properly represent canopy processes in tall, dense, or mountainous forests such as the location of this study.

Based on these new findings, further investigations are necessary. In particular, actual improvement at this study site by applying multi-layer scheme and new parameterizations and global-scale tests will be the next goal. Also, to enhance the reliability of the land surface model, more observations of water movement and energy exchange are essential at both this site and other locations in the neotropics. Tracking the spatial heterogeneity of variables related to canopy structure (e.g., leaf temperatures, leaf distributions, canopy water) is particularly important.

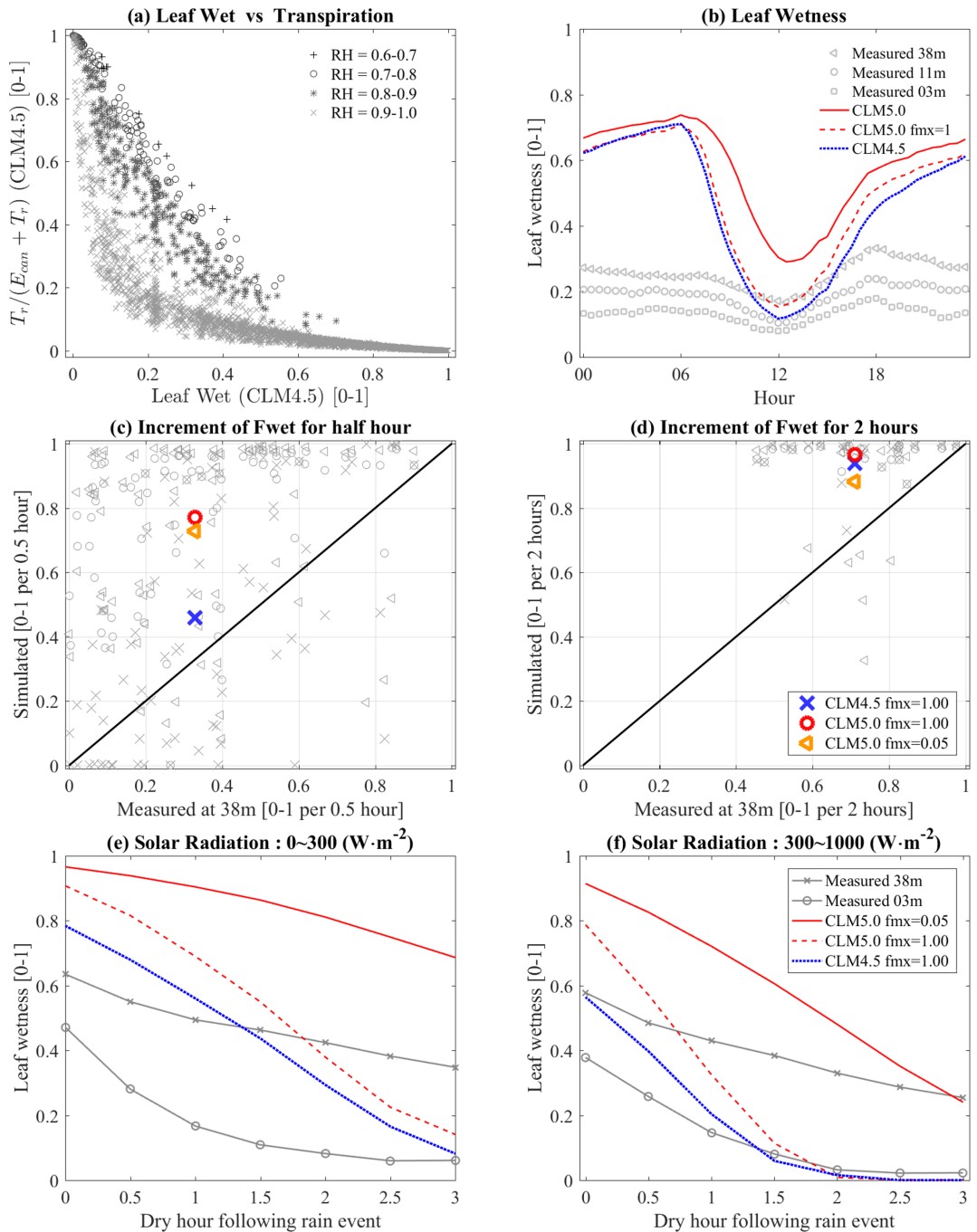

**Figure 8.** (a) TR/ET versus leaf wetness and classified by relative humidity [0-1]; (b) The ensemble diurnal variation of leaf wetness; (c) and (d) indicate interception rates; (e) and (f) are the behavior of drying canopy. The marked lines are from measurements, and lines are estimated from CLM.

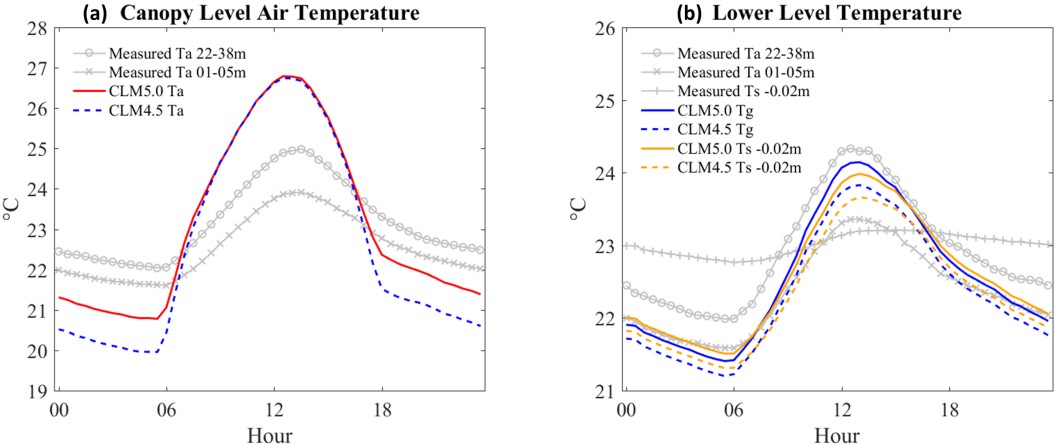

**Figure 9.** The ensemble diurnal variation of air temperatures. Canopy Levels at 22-33m and 1-5m are measured air temperatures, Ta represents air temperature at 28.075m in CLM, and Tg is ground air temperature at 0.01m in CLM. 'Ts -0.02m' is measured/simulated soil temeprature. In (a), both CLM4.5 and CLM5 is overestimated in daytime (+0.8 and +1 $C^\circ$) and underestimated during the nighttime (-1.9 and -1.1 $C^\circ$). In (b), differences between 'Measured Ta 01-05m' and all CLM values (CLM5.0 Tg, CLM4.5 Tg, CLM5.0 Ts, and CLM4.5 Ts) are -0.39, -0.14, -0.32, and -0.06 in daytime and -0.02, 0.18, -0.11, and 0.08 $C^\circ$ in nighttime. Differences with 'Measured Ts -0.02m' are -0.04, 0.21, 0.03, and 0.30 in daytime and 0.90, 1.10, 0.81, and 1 $C^\circ$ in nighttime.

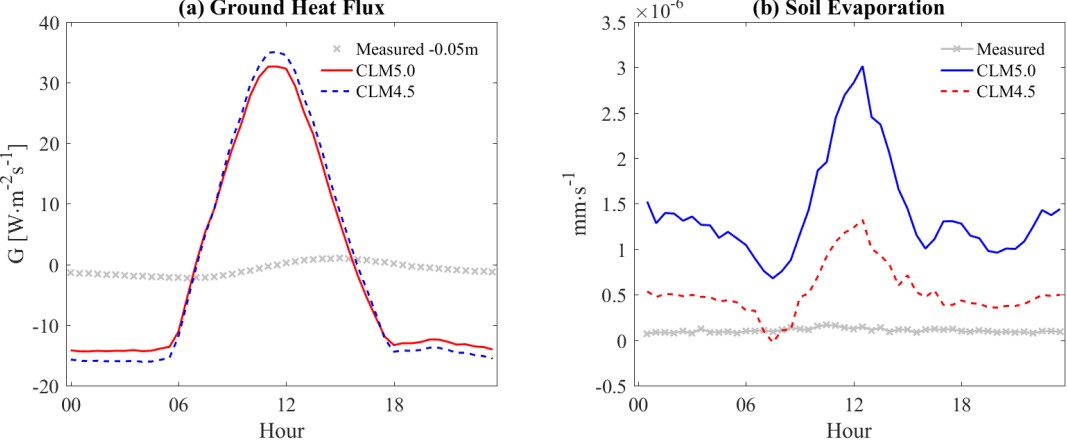

**Figure 10.** The ensemble diurnal variation of soil/ground heat fluxes (into soil +) (left) and soil evaporation.

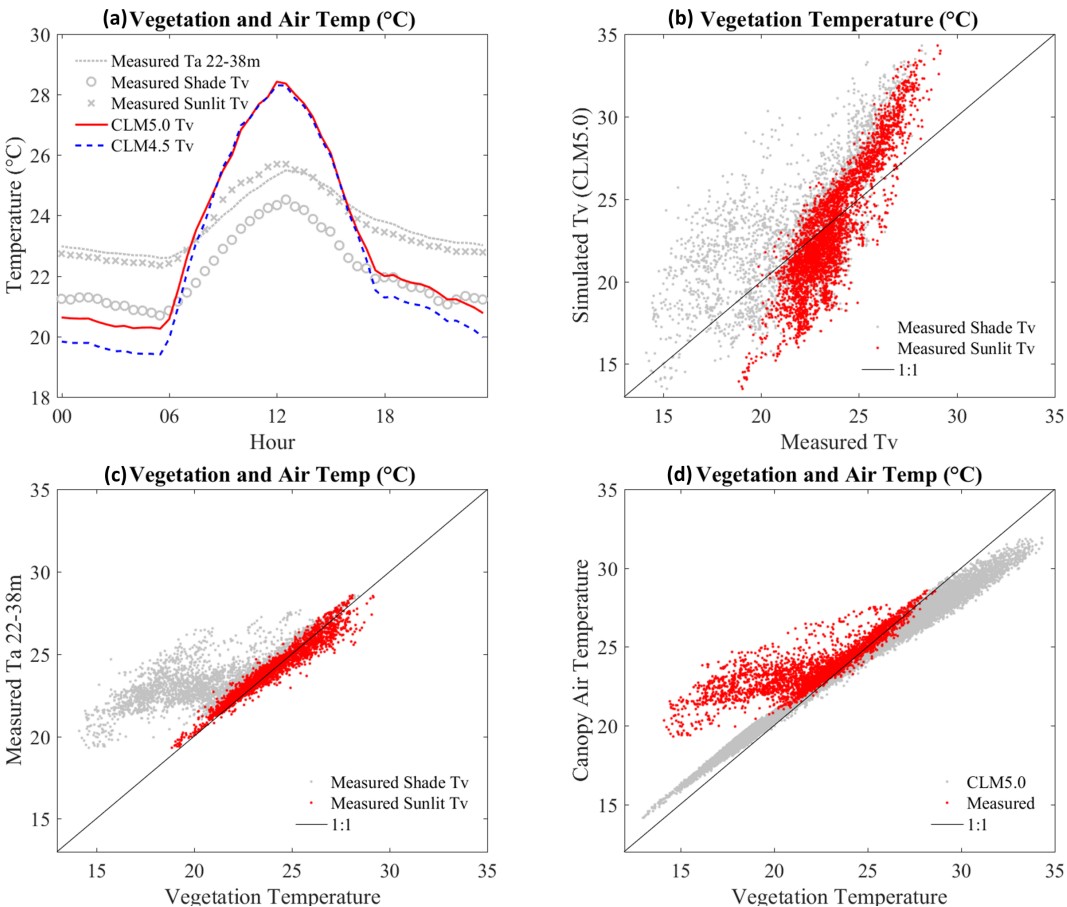

**Figure 11.** (a) The diurnal variation of leaf temperatures with measured canopy air temperatures; (b) The one to one plot of leaf temperatures: CLM vegetation temperatures (Tv) are compared with measured values for the both gauged shade (Shade Tv) and sunlit (Sunlit Tv) vegetation temperatures; (c) The one to one plots about measured canopy air temperatures versus measured leaf temperatures (sunlit and shade); (d) The one to one plots about canopy air temperatures versus leaf temperatures from CLM (CLM5 Ta vs CLM5 Tv) and observation (Canopy Ta 22-38m vs averaged Tv from sunlit and shade Tv): Averaged Tv is estimated through $(LAI_{Shade} \times Tv_{Shade} + LAI_{Sunlit} \times Tv_{Sunlit})/LAI$. In (a), daytime differences 'CLM 5.0 Tv' minus measurments ('Measured Ta 22-38m', 'Measured Shade Tv', and 'Measured Sunlit Tv') are 1.1, 2.4, and 1.0. In nighttime, the differences are -2, -0.3, and -1.8 $C^{\circ}$. CLM5 normally 0.2 higher in daytime and 0.8 higher in nighttime.

*Data availability.* All field data used in this study are archived in the OAKTrust repository (Miller et al., 2018a, b, c, d). Input forcing data for the simulation (Song, 2020) and CLM model code (Sacks, 2020) are available via Github.

## Appendix A: Estimating leaf area density (LAD) profiles based on the Beer-Lambert Law

Estimates of leaf are density were developed based on a series of Photosynthetically Active Radiation (PAR) measurements in the canopy. This site has a canopy walk-up tower situated on a steep slope between two large trees. A net radiation sensor 600 (CNR1) is located at the top (44 m), This provided incoming solar radiation ($R_s$) data which was in turn converted into PAR data (PAR = $R_s \cdot 0.5 \cdot 4.5$, where PAR is in units of $(\mu mol \cdot m^{-2} \cdot s^{-1})\cdot$ and $R_s$ is in $(W \cdot m^{-2})^{-1}$. Five PAR sensors were permanently situated at heights of 05, 21, 27, 32, and 38-m; data from these has been collected at 5-minute intervals since 2014. To complement this data, a line quantum sensor (LI-191R, LI-COR) was utilized to measure PAR at all levels over 1 to 3 hours on three sunny days: 2016/01/31, 2016/02/01, and 2016/02/04 [Figure A1]. This sensor was manually transported 605 to each tower platform, where a 1-minute integrated value was measured. This allowed us to determine PAR at equal height intervals of 1, creating a profile measurement. This data was then synced by time to the PAR data from the permanent sensors, which was temporarily collected at shorter, 1-s intervals. Weather conditions (e.g., incoming solar radiation) did not change abruptly during these field campaigns.

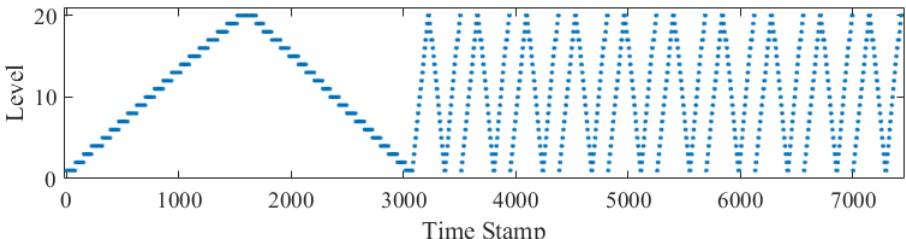

**Figure A1.** Data points collected using the line interval sensor at a 1-s sampling rate (2016/01/31 at 10:00-11:30, 2016/02/01 at 15:00-16:00, 2016/02/04 from 09:00-12:00). In the first day (the first up and down), more data points are collected at each platform level, while on subsequent visits, shorter but more frequent visits occur.

LAD was estimated based on the Beer-Lambert Law (Lalic et al., 2013; Maass et al., 1995). This was appropriate, given that 610 most LSMs follow this law to estimate radiative transfer. The Beer-Lambert Law can be written as

$$ln\left(\frac{Q_z}{Q_{max}}\right) = -\frac{k}{cos\theta} \cdot LAI(z) \tag{A1}$$

where $Q_z$ is photosynthetically active radiation (PAR) at level $z$, $Q_{max}$ is maximum PAR at uppermost location, $k$ is the canopy extinction coefficient, $LAI(z)$ is cumulative leaf area index (LAI) at $z$ level, and $\theta$ is solar zenith angle. Using the equation, leaf area index profile $LAI(z)$ can be estimated through $\theta$ and measured $Q$ which vary in time and height. If the time dependent variables are moved into the left-hand side, the relationship for $LAI(z)$ can be established via averaging the time

dependent term $E[X(t)]_t$ as

$$E\left[\left(\frac{Q_{t,z}}{Q_{t,max}}\right)^{cos\theta_t}\right]_t = e^{-k\cdot LAI(z)} = \bar{q}_z \tag{A2}$$

where $\bar{q}_z$ is normalized light extinction. In this experiment, PAR data $Q_{t,z}$ were measured using permanent sensors on the tower for long term observations, or a line quantum sensor (LQS), synced in time with the tower sensors, for high vertical resolution (1.8-m). One complication is that data measured by LQS must be normalized for across the profile, because each level cannot be simultaneously measured [Figure A2]. Therefore, continuously observed data on the top of the tower were employed as a
reference and the PAR profiles were regenerated using Eq.(A2). In the same manner, the other static PAR sensors were used to estimate light extinction data for validation of the campaign data.

| Level | H (m) | Data Available |
|---|---|---|
| Top | 44.0 | |
| 20 | 37.5 | |
| 19 | 35.7 | |
| 18 | 33.9 | |
| 17 | 32.0 | |
| 16 | 30.2 | |
| 15 | 28.4 | |
| 14 | 26.5 | |
| 13 | 24.7 | |
| 12 | 22.9 | |
| 11 | 21.0 | |
| 10 | 19.2 | |
| 9 | 17.4 | |
| 8 | 15.5 | |
| 7 | 13.7 | |
| 6 | 11.9 | |
| 5 | 10.1 | |
| 4 | 8.2 | |
| 3 | 6.4 | |
| 2 | 4.6 | |
| 1 | 2.7 | |
| | | Time flow |

**Figure A2.** Data collected at each level by the line quantum sensor (gray) and the top of canopy sensor (dark gray) for Eq.(A2) for a single campaign event (i.e., one cycle up and down the tower).

Based on this idea, Eq.(A2) for LQS is rewritten as

$$\bar{q}_z = E\left[\left(\frac{Q_{t,z}^{LQS}}{Q_{t,max}^{Tower}}\right)^{cos\theta_t}\right]_t \tag{A3}$$

where $Q_{t,z}^{LQS}$ is data measured by the LQS and or from the tower sensors $Q_{t,z}^{Tower}$ for comparison. Also, $LAI(z)$ is cumulative LAD, so LAD is written as an integrated form of

$$LAI(z) = -\int_{z_{top}}^{z} LAD(z)dz = -LAI_{tot}\int_{z_{top}}^{z} a(z)dz \tag{A4}$$

where $a(z)$ is leaf area density function $(m^2 \cdot m^{-3})$: $\int_{z_{top}}^{z} a(z)dz = 1$, $z_{top}$ is a height of the top, so $LAI(z_{top})$ refers to total $LAI$ ($LAI_{tot}$). Then, after combining Eq. (A2) and Eq. (A4), the two are rewritten as

$$\log(\bar{q}_z) = -k \cdot LAI_{tot}\int_{z_{top}}^{z} a(z)dz \tag{A5}$$

To get the density $a(z)$, it is differentiated as

$$\log(\bar{q}_z)' = k \cdot LAI_{tot} \cdot a(z) \tag{A6}$$

The numerical results are shown in Figure (A3), as $\log(\bar{q}_z)$ and $\log(\bar{q}_z)'$ to facilitate presentation. To produce the final LAD profiles, all data sets were smoothed using a spline interpolation method, and negative values were eliminated. In general,

$k \cdot LAI_{tot}$ is constant, making it a one parameter model. If $LAI_{tot}$ is known, $k$ can be estimated through Eq. (A5) and Eq. (A6).

    This analysis indicated that very short period data and simple field measurements can produce LAD profiles. The PAR profiles were measured using an LQS, and the LAD values were estimated from the equations Eq. (A5) and Eq. (A6). The profiles from LQS had a high vertical resolution over a short temporal period. The simulation results were compared with data

from several long-term sensors on the tower, and the results showed strong agreement [Figure A4].

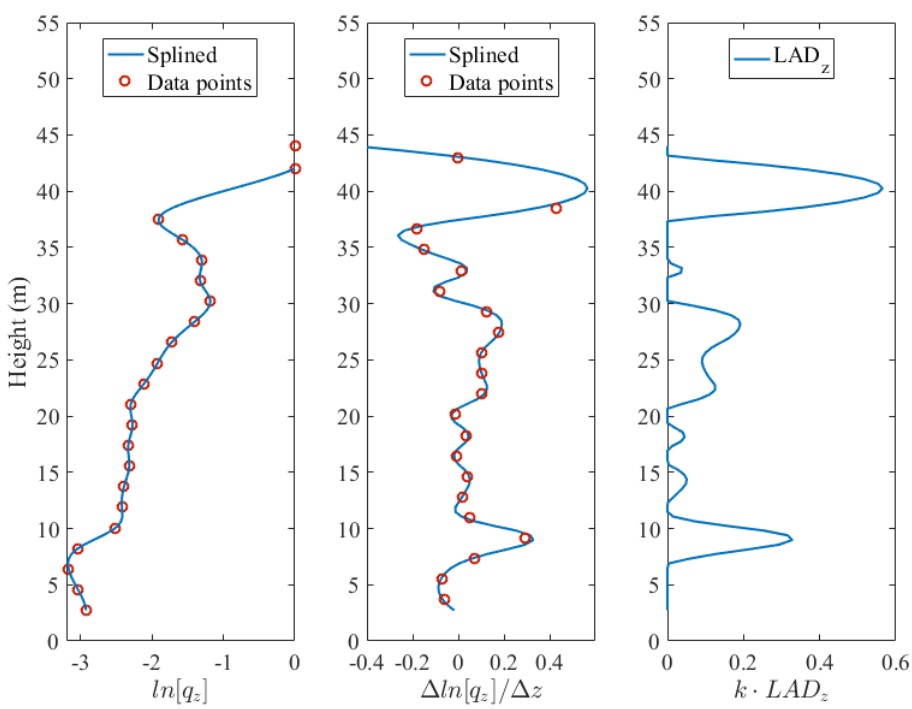

**Figure A3.** Values found at each step in calculating LAD and $LAI(z)$ profiles. Here, $ln[q_z]$ is equal to $-k \cdot LAI(z)$, and $\Delta ln[q_z]/\Delta z$ is identical to $-k \cdot LAD_z$.

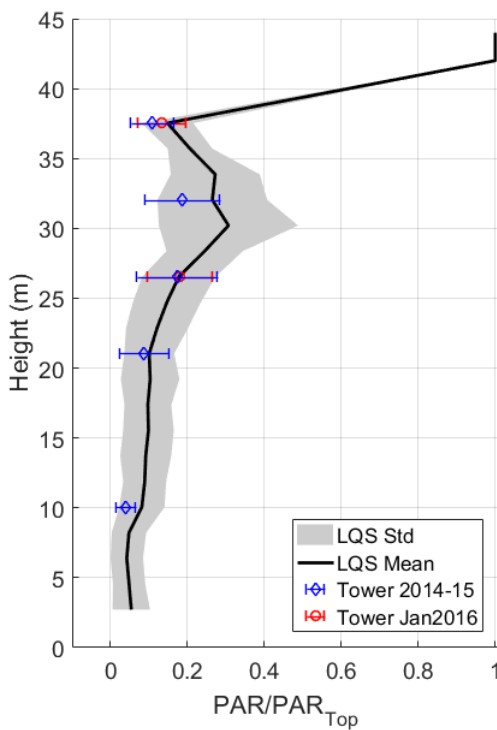

**Figure A4.** $PAR/PAR_{Top}$ profile is measured based on Eq.(A2). 'Tower 2014-15' is measured data at the tower from 2014 to 2015. 'Tower Jan2016' is measured data at the same time with LQS data.

*Author contributions.* JS performed the CLM4.5 and CLM5.0 simulation and analysis. ATC, GRM, LA, and GM installed and maintained the measurements at the site. ATC, GRM, and GM designed this project. JS wrote the manuscript with all co-authors' contributing.

*Competing interests.* There are no competing interests present.

*Acknowledgements.* This research was supported by the Office of Science (BER) U.S. Department of Energy (DE-SC0010654). The re-
searchers would like to thank the TAMU Soltis Center staff, particularly Dr. Eugenio Gonzales, Johan Rodriguez, Noylin Rodriguez, and Ronald Vargas Castro, for their logistics and maintenance support. We would also like to thank the Hammer and Soltis families for their contributions to the site's infrastructure. Also, The authors would like to thank the two anonymous reviewers and the editor for their feedback

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
