# Peer review of "Modeling land surface processes over a mountainous rainforest in Costa Rica using CLM4.5 and CLM5"

_Geoscientific Model Development, 2019_

## Referee Comment (RC1) · Anonymous Referee #1 · 15 Apr 2020

Song and colleagues present comparisons between observations and two versions of the Community Land Model in Costa Rica.

I totally agree that we need more model evaluations in the wet tropics, which is a pivotal region in the evolution of the future carbon cycle.

But for this we have to firstly compare apples with apples and we secondly need not only model evaluations but also show ways how to improve the models. For example, I think that the comparison between observed PAR and modelled PAR is a very inaccurate comparison because the PAR sensors where shaded by a nearby emergent tree while the model calculated PAR from incoming global radiation above the emergent

tree. One could have also added radiation reahing the ground, which is part of the two-stream approximation, to Figure 3e to compare with the 10 m observations. An example for the second point would be that slope and aspect could have been implemented pretty easily in CLM by simply changing the zenith angle. This does not mean a full implementation of slope and aspect in the whole land surface model as an offline and online model running on the whole globe but it would have shown a way to improve the model.

In this respect, the eddy measurements were surely far from optimal. One should then be also quite cautious in their interpretation. I was really quite worried by the repeated claim that the quantum efficiency of photosystem II should be much lower. This claim comes from simple comparison of uncertain GPP estimates with APAR values, which come either from the net radiation sensor above the canopy or from the shaded PAR sensors, which are up to 70% different (not specified in the manuscript). It should be at least surprising that the estimated GPP does not show any saturation. Instead of the quantum efficiency, also APAR could be wrong. The analysis via an apparent quantum yield neglects. for example, also sun and shaded leaves. An apparent quantum yield could be lower than the quantum efficiency because of a wrong partitioning of sun and shaded leaves, a decrease of nitrogen within the canopy that is non-exponential, wrong leaf temperatures, etc. Nothing like this is discussed in the manuscript.

For me the interesting part starts at Figure 7. I think that one can learn most about leaf wetness and model temperatures from the current data set. And it looks like that the single leaf temperature for sunlit and shaded leaves might be the main culprit of the model deficiencies. Wrong leaf temperatures lead also to erroneous canopy evaporation and hence wrong leaf wetness. The single vegetation temperature is not enough discussed in the manuscript. The literature about scaling (e.g. Wang and Leuning 1998, de Pury and Farquhar 1999) is neglected. Soil temperature, G, soil evaporation all depend on the short-wave and the long-wave radiation reaching the ground. The former could be compared to PAR at 10 m, which would give a hint if it is the

radiation scheme that needs updating or the calculation of canopy and/or canopy air temperature.

In summary, I would recommend to refocus the manuscript to the temperatures and leaf wetness. If you provide ideas how to improve the model, the manuscript might fit to GMD. At the moment, the manuscript matches rather the scope of Biogeosciences. The latter would also offer the possibility to highlight more the unique observations. They are much more criticizable in the context of a model comparison.

Some specific remarks that I noted during reading the manuscript:

- The introduction reads like a defense why we need model-data comparisons in the wet tropics. This is more than obvious to me.

- I could not access the PhD thesis Song (2019), while I would have been interested to know how he determined LAI.

- I could not find the figure that show that the "predominant winds flow parallel to the valley (e.g., N-S) and not perpendicular to the mountain slope." (line 132f). Why e.g.?

- Line 170ff has already opinions about model formulations in the method section.

- A 100 year spinup? This is much too long for energy and water and not enough for carbon.

- There is often the mentioning of "oversimplification". Is a process that is not implemented in a model an oversimplification?

- Line 362-375 is gibberish. I did not understand the sentences.

[Figure]

- Section 3.5: I think that the formulation "the simulated temperature might be overly sensitive to incoming solar radiation" is unphysical to say the least. Be more specific, more process-related.

- Line 425: "This study demonstrates the possibility of reducing predictive uncertainty by adapting the model to mimic such slope effect ..." The study did not show this. It only demonstrated that one can improve comparison by reducing the quantum efficiency. This is not mimicking a slope effect.

- Line 508ff mentions a good point and this should be elaborated. How could this be improved? Should there be different wetness fractions for sunlit and for shaded leaves? How would you implement this? If light changes. i.e. the fraction of sunlit and shaded leaves change as well, what would you do with the excess (or missing) water that come from purely changing the fractions without any evaporation or percolation yet? What other models would be less "physically simplified"? The Gash model? The Rutter model?

---

## Referee Comment (RC2) · Anonymous Referee #2 · 11 Jun 2020

**General comments**

In the presented manuscript, the authors provide a comprehensive assessment of the performance of the two latest versions of the Community Land Model (CLM4.5 and CLM5.0) at a tropical montane forest in Costa Rica. A broad range of measurements are available at the chosen location including radiation fluxes, $CO_2$ fluxes, water vapor fluxes, leaf wetness, temperatures at different locations in and around the canopy, and the ground heat flux. The authors identify a number of discrepancies between the field measurements and the two model versions including an over-estimation of the surface albedo, the gross primary productivity, ET, the leaf wetness, and the diurnal variability of temperature. Also, they demonstrate that the overestimation of the gross primary productivity by the model could be alleviated be choosing a lower value for the quantum efficiency of photosystem II than the default value. Further, decreasing the maximum fraction of wet leaves in CLM5.0 reduced the overestimation of ET.

Overall, studies such as the one presented here provide valuable insights for further developing the model and I could learn a lot about the model from reading the study. Therefore, I think the manuscript is definitely worth publication. However, it was sometimes hard to follow. A lot of the detailed comments below address such issue and are hopefully helpful in increasing the readability of the text. Also, I wonder whether the model was challenged with an unfair comparison on some aspects:

-ET/TR: If I interpret Fig. 6 c correctly, the average TR from the sapflow measurements is as large or even larger than the average ET from the EC measurements (integrated over the entire day). This would imply that either the sapflow measurements overestimate TR (because the sampled trees are not representative? The setup described in Aparecido et al., 2016 is convincing though.) or the EC measurements underestimate ET (because the EC method is problematic on sloped terrain?), as one would expect that ET is higher than TR at a site with considerable interception by leaves. In fact, the simulated TR of the CLM versions seems to be quite realistic. Also, I wonder whether it makes sense to exclude nighttime water fluxes from the analysis with the argument that CLM does not represent nighttime TR. As the authors mention in lines 240-245, the sapflow measurements exhibit a temporal lag, where part of the daytime TR originates from plant water uptake during the night. Wouldn't it make more sense to compare values integrated over the entire day for a fairer comparison? I agree that the diurnal cycle of ET is relevant and should ideally be captured by the model. But still as a starting point, a good representation of the daily average value is already important.

-Leaf wetness: I am not sure whether I understood the normalization correctly. 0 corresponds to complete dryness of the leaves and 1 for fwet = fwetmax in the respective model configuration? If this is the case the actual maximum in the diurnal cycle of the leaf wetness in CLM5 would be ~0.7*0.05 = 0.035. This would mean that CLM5 vastly underestimates the leaf wetness compared to the measurements rather than overestimate it as Fig. 7 suggests. Also why did you not test for an intermediate fwetmax (e.g., 0.5)?

**Specific comments**

L17: I am not sure what the authors mean by climate cycles.

L24: Greater energy exchange than what?

L105: A brief statement about the seasonality could be of interest to the reader here.

L108: The base of which mountain. Providing a map with the location of the two sites might help the reader to get a clearer picture of the field sites.

L131: 33 or 34 m? In Fig. 1 the EC is located at a height of 33 m. Also what does IRGA stand for?

L158-160: Soil moisture could still limit stomatal conductance in the model. However, it is probably a fine choice to neglect soil moisture limitation in this study, as ET and the carbon fluxes are on the high side in the model.

L260: Are you sure you are talking of the canopy air temperature here and not the 2 m temperature? The canopy temperature Ts has a different definition if I understand correctly (see eq. 5.93 in technical documentation of CLM5.0).

L286-287: Albedo cannot explain the difference in the nighttime net radiation. Also, how is the albedo estimated?

L307: To me it is unclear what is meant by the BB parameter.

Fig. 4: Are the APAR values in panels c and d from the observations or the model? Part of the discrepancy in the alphas could also originate from differences in PAR (Fig. 3 f).

L344-345: The $R^2$ increased marginally for CLM5 when introducing phi=0.25. But, the slope and intercept is clearly deteriorated by this modification. So, I wouldn't really talk of an improvement here.

L390: Another relevant process could be energy storage in particular in the stem, which is missing in the default version of CLM5. This process was found to decrease the diurnal temperature range in forests and decrease the overestimation of turbulent heat fluxes in CLM5 (Swenson et al., 2019, Meier et al., 2019).

L433: A nice study supporting and concretizing your claim for accounting for terrain effects: Fan et al. (2019).

**Technical comments**

L19: I assume 16 % of the land area right? Also, I would reformulate ", accounting for 33 %" to "and account for 33 %".

L46: Remove balance after water.

L49-51: This sentence reads a bit awkward for me. Please consider reformulating it.

L55: Remove on before "tropical ET"

L59: Verb missing. Maybe "which implies that net radiation is highly…" or "which implies a correlation between net radiation and …"

L72: Remove "such" before complicated systems.

L82: The sentence "Therefore, site-based…" breaks the flow of the text for me. Consider removing it.

L95: Parameter choices instead of parameters?

Caption Fig. 1: Add space at "points(100 …"

L126: Remove canopy after height.

L132: As shown in which figure? Again you could show the mean wind direction nicely in a map.

L141: Consider removing "Additionally".

L146: Consider reformulating "models determining the amount of energy exchange" to "models to determine the exchange of energy".

L146-148: This sentence reads awkward for me. Please consider reformulating it.

L148: Remove "energy of"

L149: Consider rearranging this sentence to "For example the canopy energy balance can be written as a function of vegetation temperature:"

L155: Consider formulation "Using a big-leaf approach, CLM represents…"

L156: Shade**d** leaves.

L166-169: I was confused, when I read this for the first time. Consider to first mention CLM4.5 and its equation and then CLM5 and its equation.

L174: 0.1 has the unit $kg/m^2$ right?

L181: Consider reformulating to "...and the $CO_2$ concentration does not vary significantly."

L186: Is there an alternative way to estimate the electron transport rate in the model? If not, remove "additionally".

L188: = 0.7 **by** default)
Also consider to finish the sentence after "electron transport" and adding a "where" before the phi.

L195-197: I am confused about the units here. $c_i$ and $c_p$ are supposed to be concentrations right? Also consider restructuring these sentences as it reads a bit awkward.

L211: Consider changing "model if" to "model, as".

L212: "include" instead of "included".

L214: Maximum fractional saturated area of what? Also, add "and" before color class.

L238: I do not understand what the authors mean by "whether the model is over-parameterized".

L239-240: Consider reformulating "To investigate water loss from the canopy".

L249-250: Consider reformulating this sentence.

L251: by a combination **of** the Medlyn …

L257: Consider changing to active form: "it is necessary…" to "CLM requires a number of heights  as input variables.

L268-269: Consider splitting into two sentences: "… soil layers. Therefore, the results …".

L271: Consider reformulating to "cycling (through) the 6-year forcing data collected…".

L279: Consider avoiding passive voice in last part of the sentence.

L284: Replace "CLM" with "Net radiation in CLM".

L292: Remove extra dot.

L292-293: Consider reformulating to ", when the albedo difference between the observations and the simulation was smallest (+0.0214)."

L294: Add comma after "albedo models".

L303: Are you talking of optical thickness here?

L308: Consider reformulating to "Disabling the plant hydraulic stress model in/of CLM5…".

L311: possible causes of **the** discrepancy…

L315: Add comma after 0.05.

Fig. 3: I am missing a unit in panels d and f.

L332: I would suggest to just refer to Fig. 6 here.

L341: Refer to relevant figures here.

L342: Consider reformulating "and reduced change to the estimated" to "reduction of".

L372: Consider replacing "assume" with "play".

L380: Not sure if I understand this correctly. 3 hours of no rain would require 6 consecutive half-hourly time steps with no rain.

L414: This should read "lower in the measurements".

L428: Studying…

L434: The formulation "are becoming more elaborate vertically and horizontally" sounds a bit awkward to me. Consider reformulating.

L439: Consider splitting into two sentences: "...2012). Such errors…".

L450-452: Consider reformulating to "Unlike the $CO_2$ fluxes, which largely depend on plant-light relationships and their effect on photosynthesis, ET consists of three major components: …". Also, the $CO_2$ fluxes consist of multiple components as well (i.e., photosynthesis, soil respiration, plant respiration). Therefore, consider removing the comparison to the $CO_2$ fluxes completely.

L480: Consider replacing "datasets" with "counterparts".

L481: related **to** each other … canopy condition**. Therefore,** it was (the latter modification being a suggestion).

L483: during **the** daytime.

L496: Remove "with".

L508: Consider replacing "seems too physically simplified" with "seems oversimplified".

L524: Remove "balance)"

L535: Consider replacing "such as this" with "such as the location of this study".

Fig. 7: Shouldn't the unit in panels c and d be per time? Otherwise, I misunderstand these panels.

*References*

*Fan, Y., Clark, M., Lawrence, D. M., Swenson, S., Band, L. E., Brantley, S. L., et al. ( 2019). Hillslope hydrology in global change research and Earth system modeling. Water Resources Research, 55, 1737 1772. https://doi.org/10.1029/2018WR023903*

*Meier, R., Davin, E. L., Swenson, S. C., Lawrence, D.M. & Schwaab, J. (2019). Biomass heat storage dampens diurnal temperature variations in forests. Environ. Res. Lett. 14, 084026, DOI: 10.1088/1748-9326/ab2b4e.*

*Swenson, S., Burns, S. P., & Lawrence, D. M. ( 2019). The impact of biomass heat storage on the canopy energy balance and atmospheric stability in the community land model, Journal of Advances in Modeling Earth Systems, 11, 83 98. https://doi.org/10.1029/2018MS001476*

---

## Author Comment (AC1) · 17 Jul 2020

**[Cover Letter]**

Dear Reviewers,

Thank you for the opportunity to respond to comments on our manuscript. We are grateful to the reviewers that have reviewed our manuscript and provided us with valuable feedback. Your insightful comments lead to a number of improvements in the current version. We considered them carefully and did our best to appropriately address them. We welcome your continued input.

Sincerely,
Jaeyoung Song, Ph.D., Postdoctoral Researcher
Department of Civil and Environmental Engineering
Texas A&M University

**Reply to Reviewer 1**

**[General Comment 1]** Song and colleagues present comparisons between observations and two versions of the Community Land Model in Costa Rica. I totally agree that we need more model evaluations in the wet tropics, which is a pivotal region in the evolution of the future carbon cycle.
**(a)** But for this we have to firstly compare apples with apples and we secondly need not only model evaluations but also show ways how to improve the models. For example, I think that the comparison between observed PAR and modelled PAR is a very inaccurate comparison because the PAR sensors where shaded by a nearby emergent tree while the model calculated PAR from incoming global radiation above the emergent tree. **(b)** One could have also added radiation reading the ground, which is part of the two-stream approximation, to Figure 3e to compare with the 10 m observations. **(c)** An example for the second point would be that slope and aspect could have been implemented pretty easily in CLM by simply changing the zenith angle. This does not mean a full implementation of slope and aspect in the whole land surface model as an offline and online model running on the whole globe but it would have shown a way to improve the model.

**Response to General Comment 1:** We take the reviewers point that it can be difficult to properly compare the single-layer model with existing point-scale datasets. The single-layer model cannot provide outputs to plausibly compare with profile data like PAR and leaf wetness, especially for such a complex site. We attempted to make as direct of a comparison as possible and disagree that the values we selected were unfair; we have added some text, as shown below, to help clarify.

**Response to General Comment 1(a):** In Figure 4e (formerly Figure 3e), we show only those PAR values that are associated with the shaded fraction, as described on L309 (formerly L297): "The measured PAR values, generated by sensors somewhat shaded by the upper canopy, were diurnally skewed compared with shaded PAR from CLM." For this comparison, we do not include the sunlit PAR calculated by CLM or the top-most observation (44 m). At this site, the 38 m sensor can be sunlit for a portion of the day (e.g., in the morning due to the sloping surface), so we cannot precisely define it as a shaded location. Hence, we provided diurnal-variation plots for all possible shaded PAR data. We revised the manuscript, adding the text below:

1. L332 - "The APAR, including sunlit and shade leaf area, was estimated in CLM using measured incoming solar radiation above the canopy at 44 m."

**Response to General Comment 1(b):** CLM does not calculate ground PAR values. We could add it in Figure 4e (it was Figure 3e) by estimation, but the ratio of the absorbed energy was provided at L533 (formerly L491): "The ratio of the absorbed energy on the soil surface to the total incoming solar radiation in CLM was 0.03, but our PAR profile data [Figure 4e] indicated the ratio should be lower, around 0.01". Also, we have ground flux comparison too [Figure 10].  To clarify, we added:

2. L534 - "The average incoming solar radiation in the daytime was around 300 $W/m^2$. Estimated absorbed energy on ground and vegetation in CLM and the received energy at 10 m PAR sensor (unit was converted) were 9.4, 252.5, and 3.1 $W/m^2$."

**Response to General Comment 1(c):** As suggested, we modified the radiative transfer models by applying different angles. It showed a slight difference, but also resulted in other errors (e.g., skewed PAR to the opposite side). This is mainly because incoming radiation consists of direct and indirect radiation. The slope effect is primarily related to the direct radiation. However, diffusive radiation (indirect) also occupies quite a significant portion. Hence, the modification of only direct radiation did not give dramatic improvement. We updated:

3. L317 - "This feature can be important because the hill-slope surface is more sensitive to sun angles. It can affect to determine the sunlit/shaded area. Simple manipulation was attempted by changing the solar angle to mimic the slope effect on albedo [Figure 4c; Figure4d; Figure 4e]. The cosine zenith was re-estimated by pushing back 30 degrees, to apply to the light extinction coefficient K in two-stream approximation. This simple modification reduced some the skewness of albedo [Figure 4d]. However, shaded PAR showed opposite behavior compared to the observation, mainly because sunlit area was increased."

4. Figure 4c,d,e were updated.

5. L446 - "A simple modification to albedo was attempted but it requires more complicated manipulation to match variables other than albedo (e.g., PAR). This finding suggests multiple layer scheme is necessary to properly represent light penetration."

**Finally**, we have tested different photosynthesis models (Leuning model (Leuning, 1995), WUE (Katul et al., 2010)), leaf wetness models (Aston, 1979), and solar angle parameters as above. We just briefly mentioned some tests in the context. The problem was that these updates did not provide significant improvement or caused other issues. We have decided to improve the model structure first before deeply studying the update of parameters. We have updated and tested the multi-layered CLM, which provided a significant improvement, and this is the subject of our next paper. Hence, we decided to submit this manuscript as a model evaluation paper, which we considered to be the first step in this ongoing process. We believe that adding some information briefly about what we have tested can resolve this issue.

[**Major Comment 1**] **(a)** In this respect, the eddy measurements were surely far from optimal. One should then be also quite cautious in their interpretation. **(b)** I was really quite worried by the repeated claim that the quantum efficiency of photosystem II should be much lower. This claim comes from simple comparison of uncertain GPP estimates with APAR values, which come either from the net radiation sensor above the canopy or from the shaded PAR sensors, which are up to 70% different (not specified in the manuscript). It should be at least surprising that the estimated GPP does not show any saturation. Instead of the quantum efficiency, also APAR could be wrong. The analysis via an apparent quantum yield neglects. for example, also sun and shaded leaves. An apparent quantum yield could be lower than the quantum efficiency because of a wrong partitioning of sun and shaded leaves, a decrease of nitrogen

within the canopy that is non-exponential, wrong leaf temperatures, etc. Nothing like this is discussed in the manuscript.

**Response to Major Comment 1(a):** Yes, the eddy-covariance measurements are not optimal, as mentioned in this manuscript. Although the system was located above the canopy and had some distance from the emergent tree (because of the steep surface), but we could not sure how well the sensor represents the site's fluxes. We believe that neither the model or the data should be considered as "truth", but that they can both provide valuable insights, and have attempted to convey that uncertainty in the paper. We also found a partial solution through a multi-layered scheme mentioned above.

**Response to Major Comment 1(b):** About the quantum efficiency GPP appears saturation [Figure R1]. Also, a two-stream approximation method estimated APAR through CLM, which is used widely in our community. Hence, I believe that APAR estimation should be reasonable although [Figure 4e; Figure 4f] indicates there is a possible error determining sunlit and shaded PAR. Here, the actual APAR value of this field should be higher than the flat forest by larger sunlit area, because a canopy is placed in semi-open space due to the slope effect. In this circumstance, the quantum yield and the quantum efficiency should be lower. As the reviewer pointed out, the estimation of APAR cannot be sufficient because default CLM does not take or provides a spatial radiation profile. We also mentioned this could not be an exact solution at L343 (previously L320) – "For this study, $\Phi$ was modified to get proper $\alpha$, but the issue should be revisited in future studies.". Also, we revised this manuscript as below:

6. L340 - "Of course, this analysis itself has a possible error by the eddy-covariance measurement and APAR estimation. APAR, which estimated by CLM, contains only sunlit and shade leaf area, making it too simplistic."

[Figure]

*Figure R1:GPP box plot from the observation. The initial slope was reestimated using data belonging to the initial part. The slope parameter is 0.026, which is slightly higher. Also, we can recognize GPP saturation.*

The reviewer's comment that "An apparent quantum yield could be lower than the quantum efficiency" is an excellent point for discussion. However, we don't have a direct measurement to compare these two. Instead, we derived equation to connect between the apparent quantum yield and the quantum efficiency. Hence, this study was not for comparison between apparent quantum yield and quantum efficiency.

Instead, this study has an assumption that apparent quantum yield was estimated through $\partial J_x/\partial I_{APAR}\times0.667\times0.25$ at $I_{APAR}=0$. Also, we revised this manuscript as below:

7. L206 - "It is worth noting that the differential has brought independence from $J_{max}$ at zero APAR, which is highly related to nitrogen and leaf temperature."

[Figure]

*Figure R2: Light-limit curve with different $J_{c,max25}$. In this study, except for the influence of daylength, $j_{c,max25}$ is approximately 59 μmol m$^{-2}$ s$^{-1}$.*

We agree that the error can be caused by the incorrect partitioning of sun and shaded leaves, a decrease of nitrogen within the canopy, or incorrect leaf temperatures. However, the effect on this parameter's change was not found to be significant in this study and some are already discussed in a previous study [Bonan 2011]. First, APAR was already partitioned as sunlit and shade leaf in the model in CLM. At this point, we believe that further structure's improvement via a multi-layered model is necessary for full partitioning. Here, the `model-layered model' means not for radiative transfer but a full energy-mass balance scheme (e.g., CLM-ml). Second, as we can see in [Figure R2; Figure 5a], the change of $J_{c,max25}$ tends to limit the maximum rate but does not affect the initial slope in CLM. The leaf temperature and nitrogen parameter (e.g., by LUNA-scaling model or BGC) affect $V_{c,max25}$, and $J_{c,max25}$., not a quantum efficiency. Here, $V_{c,max25}$, and $J_{c,max25}$ are proportionally related. The scaling and decrease of nitrogen within the canopy were also discussed using a multi-layer scheme in [Bonan 2011], and the updated model still has errors in equatorial region. Last, shade and sunlit leaf temperatures did not show a significant difference as much as affecting $J_{c,max25}$ [Figure 11a]. The sensitive parameter to change the initial slope was quantum efficiency and curvature parameter. We added:

8. L473 - "The analysis contains possible errors caused by the simplified model for APAR and measurement error for GPP."

[Major Comment 2] For me the interesting part starts at Figure 7. I think that one can learn most about leaf wetness and model temperatures from the current data set. And it looks like that the single leaf temperature for sunlit and shaded leaves might be the main culprit of the model deficiencies. Wrong leaf temperatures lead also to erroneous canopy evaporation and hence wrong leaf wetness. The single vegetation temperature is not enough discussed in the manuscript. The literature about scaling (e.g. Wang and Leuning 1998, de Pury and Farquhar 1999) is neglected. Soil temperature, G, soil evaporation all

depend on the short-wave and the long-wave radiation reaching the ground. The former could be compared to PAR at 10 m, which would give a hint if it is the radiation scheme that needs updating or the calculation of canopy and/or canopy air temperature.

**Response to Major Comment 2:** The small difference between sunlit and shade leaf temperature, as shown in the observation, may not give a notable change of leaf wetness. What we expect here is to apply a multi-layer scheme for profiled leaf wetness, including the vertical energy and water exchange schemes, which would provide many degrees of difference in leaf temperature and the leaf wetness. Since the measurements also have a large scale, we cannot identify the leaf temperature for all spots. We modified some context based on reviewer's suggestion as:

9. L493 - "We have tested more complicated interception models (e.g., Aston 1979), but they produced only a small difference in the leaf wetness."
10. L495 - "Our observations also showed variations in behavior based on height within the canopy, and such changes imply that more layers are necessary for accurate predictions of canopy water storage."
11. L553 - "Maybe, the sunlit area should intercept the precipitation first, and dry out faster than the shaded area. On the other hand, this two-layer scheme still involves up-scaling issues to embrace in-canopy variability such as the vertical segmentation of the light, physiological parameters, and the energy exchange (Bonan et al., 2011; Wang and Leuning, 1998; De Pury and Farquhar, 1999)."
12. L557 - new paragraph was added. It reads, "Vegetation temperature affected energy flux via its relationship to canopy air temperature (Ta) and physiological processes such as transpiration (Wang et al., 2014). The problem of skin (surface/leaf) temperature appeared in this study as in other reports (Wang et al., 2014; Chen et al., 2010; Zheng et al., 2012; Zeng et al., 2012). Some researchers have attributed these issues to incorrect parameterization of roughness length for heat and have made a number of advances toward reducing these errors (Yang et al., 2002; Wang et al., 2014; Chen et al., 2010; Zheng et al., 2012; Zeng et al., 2012). However, we noted that our case is different since most studies discussed low diurnal variations and underestimations. The one-to-one comparisons between the canopy air temperature and the leaf surface temperature [Figure 11c; Figure 11d] indicated that Tv on sunlit leaves normally followed the canopy air temperature (i.e., leaf thermoregulation), as described in other literature (Michaletz et al.,2016). However, CLM does not consider such leaf thermoregulation processes."

[General Comment 2] In summary, I would recommend to refocus the manuscript to the temperatures and leaf wetness. If you provide ideas how to improve the model, the manuscript might fit to GMD. At the moment, the manuscript matches rather the scope of Biogeosciences. The latter would also offer the possibility to highlight more the unique observations. They are much more criticizable in the context of a model comparison.

**Response to General Comment 2:** We respectfully disagree, as this manuscript provided 12 figures for temperature and leaf wetness, which occupy about 30% of our results. Focusing on these two variables could be problematic, because they are so heavily entwined with carbon/vapor fluxes, as described in this manuscript. We originally attempted to demonstrate additional improvements to the model here, however, we found that it made the manuscript far too lengthy. As such, we decided to submit this manuscript as a model evaluation paper and the other part as model improvement paper separately. On the other hand, we do see the rationale behind the recommendation. As a compromise solution, we added a discussion of slope effect to improve the model. The next set of work would require a multi-layer scheme (i.e., the subject of the second paper), so we provided more details on it in the discussion session.

**[Some specific remarks]:**

**S1.** The introduction reads like a defense why we need model-data comparisons in the wet tropics. This is more than obvious to me.

**Response:** In the introduction, we attempted to contextualize our work for both the LSM/ESM community as well as tropical system researchers who may be less familiar with the modeling deficits. We added a paragraph in the introduction based on Reviewer 2's suggestion at L72. If there are other specific areas we should include in our background, we would appreciate additional feedback.

**S2.** I could not access the PhD thesis Song (2019), while I would have been interested to know how he determined LAI.

**Response:** A brief explanation was at L128, and also added to Figure 2. Moreover, we newly added the detailed information in Appendix A.

**S3.** I could not find the figure that show that the "predominant winds flow parallel to the valley (e.g., N-S) and not perpendicular to the mountain slope." (line 132f). Why e.g.?

**Response:** We have eliminated "e.g." and modified the figure by inserting "E-W" to indicate directionality.

**S4.** Line 170ff has already opinions about model formulations in the method section.

**Response:** We moved it to the discussion section (L486).

**S5.** A 100 year spinup? This is much too long for energy and water and not enough for carbon.

**Response:** Yes. It should be 30 years, but it ran more just in case. This simulation was not conducted using the BGC module of CLM, so longer spinup for carbon is not necessary.

**S6.** There is often the mentioning of "oversimplification". Is a process that is not implemented in a model an oversimplification?

**Response:** Yes, we agree that all models are oversimplifications of reality. However, we use the term in the relative sense. For instance, a single-layer model is very simplified compared to a multi-layer scheme. Also, a sub-model, such as the interception model, exists in CLM, but the process is too simple. In these cases, we also used "oversimplified" when this simple model causes an error. We have adjusted the language throughout to clarify, using the terms "relatively oversimplified" and "too simplistic" as replacements.

**S7.** Line 362-375 is gibberish. I did not understand the sentences.

**Response:** L385: Updated. "Intercepted precipitation was usually too high in CLM compared to observed leaf wetness [Figure 8c; Figure 8d]. The values in [Figure 8c] and [Figure 8d] were the increasing rate of leaf wetness due to precipitation. large and thick markers indicate the average of values. The collected data was conditioned upon the absence of a rainfall event at least 2 hours prior and an initial leaf wetness lower than 0.2. [Figure 8c] shows 0.5-hour rainfall events (one consecutive event in 30-min scale) and [Figure 8d] is for 2 hours rainfall events (four consecutive events). This increment was directly related to canopy interception: the usual increment for 2-hour (and 30-min) rain was 0.71(0.33) at a 38 m height data, 0.48(0.28) at a 3 m height data, around 0.88(0.73) in CLM5, 0.97(0.77) in CLM5 fmx=1, and 0.94 (0.46) in CLM4.5. The modified interception model (CLM5 fmx=1) from Eq. (3) resulted in higher interception rate than CLM4.5 fmx=1 [Eq. (2)]. The interception rate also seemed higher with CLM5 fmx=1 than with original CLM5 as in [Figure 8c] because CLM5 fmx=1 had a higher canopy evaporation rate. This effect resulted in the acceleration of canopy evaporation while allowing interception to play a larger role in the canopy water balance. In the one-to-one comparison, the increase of leaf wetness in CLM was usually higher than in measured data. Consequently, the wet canopy fraction at the beginning of the drying process was usually higher in CLM than in the measurements: 0.63 at the 38 m observation,

0.47 at the 3 m observation, 0.96 in CLM5, 0.9 in CLM5 fmx=1, and 0.78 in CLM4.5 (see y-axis data at x-axis in [Figure 7e]).”

**S8.** Section 3.5: I think that the formulation “the simulated temperature might be overly sensitive to incoming solar radiation” is unphysical to say the least. Be more specific, more process-related.
**Response:** Added some explanation L411: “In other words, the simulated temperature may be overly sensitive to incoming solar radiation, like leaf wetness, which is likely given that overestimation and underestimation cycle followed the solar cycle.”

**S9.** Line 425: “This study demonstrates the possibility of reducing predictive uncertainty by adapting the model to mimic such slope effect ...” The study did not show this. It only demonstrated that one can improve comparison by reducing the quantum efficiency. This is not mimicking a slope effect.
**Response:** We changed it into “The study found that slope affected various data and outputs to an important degree” at L451.

**S10.** Line 508ff mentions a good point and this should be elaborated. How could this be improved? Should there be different wetness fractions for sunlit and for shaded leaves? How would you implement this? If light changes. i.e. the fraction of sunlit and shaded leaves change as well, what would you do with the excess (or missing) water that come from purely changing the fractions without any evaporation or percolation yet? What other models would be less “physically simplified”? The Gash model? The Rutter model?
**Response:** We elaborated this part (L553) and added a paragraph at L557 (see #11 and #12 in **Response to Major Comment 2**)

**[Reference]**
Andrews, R., Song, J., Miller, G. R., Cahill, A. T., W., G., and Moore: Micrometerological profiles in a lower montane tropical forest: Variations induced by diurnal cycle, leaf wetness, and seasonality, In preparation for submission to Agricultural and Forest Meteorology, 2019.

Aston, A.: Rainfall interception by eight small trees, Journal of Hydrology, 42, 383 – 396, ttps://doi.org/https://doi.org/10.1016/0022-1694(79)90057-X, 1979.

Bonan, G. B., Lawrence, P. J., Oleson, K. W., Levis, S., Jung, M., Reichstein, M., Lawrence, D. M., and Swenson, S. C.: Improving canopy processes in the Community Land Model version 4 (CLM4) using global flux fields empirically inferred from FLUXNET data, Journal of Geophysical Research: Biogeosciences, 116, https://doi.org/10.1029/2010JG001593, 2011.

Katul, G., Manzoni, S., Palmroth, S., and Oren, R.: A stomatal optimization theory to describe the effects of atmospheric CO2 on leaf photosynthesis and transpiration, Annals of botany, 105, 431–442, https://doi.org/10.1093/aob/mcp292, 2010.

Leuning, R.: A critical appraisal of a combined stomatal-photosynthesis model for C3 plants, Plant Cell Environ, 18, 339–355, 1995.

Wang, Y. and Leuning, R.: A two-leaf model for canopy conductance, photosynthesis and partitioning of available energy I :: Model description and comparison with a multi-layered model, Agricultural and Forest Meteorology, 91, 89 – 111, https://doi.org/https://doi.org/10.1016/S0168-1923(98)00061-6, 1998.

Vose, J. M., Clinton, B. D., Sullivan, N. H., and Bolstad, P. V.: Vertical leaf area distribution, light transmittance, and application of the Beer-Lambert law in four mature hardwood stands in the southern Appalachians, Canadian Journal of Forest Research, 25, 1036–1043, 1995.

**Reply to Reviewer 2**

[General Comment 1] In the presented manuscript, the authors provide a comprehensive assessment of the performance of the two latest versions of the Community Land Model (CLM4.5 and CLM5.0) at a tropical montane forest in Costa Rica. A broad range of measurements are available at the chosen location including radiation fluxes, $CO_2$ fluxes, water vapor fluxes, leaf wetness, temperatures at different locations in and around the canopy, and the ground heat flux. The authors identify a number of discrepancies between the field measurements and the two model versions including an over-estimation of the surface albedo, the gross primary productivity, ET, the leaf wetness, and the diurnal variability of temperature. Also, they demonstrate that the overestimation of the gross primary productivity by the model could be alleviated be choosing a lower value for the quantum efficiency of photosystem II than the default value. Further, decreasing the maximum fraction of wet leaves in CLM5.0 reduced the overestimation of ET.

Overall, studies such as the one presented here provide valuable insights for further developing the model and I could learn a lot about the model from reading the study. Therefore, I think the manuscript is definitely worth publication. However, it was sometimes hard to follow. A lot of the detailed comments below address such issue and are hopefully helpful in increasing the readability of the text. Also, I wonder whether the model was challenged with an unfair comparison on some aspects:

**Response to General Comment 1:**
The reviewer summarizes the main point which authors want to deliver through this manuscript and gave constructive comments to improve its delivery and structure. We installed the measurement carefully and used the data for the fairest comparison possible. However, different observation such as ET/TR can provide unexpected output due to measurement error but also different methods and scales. We believe that these issues and the unfair comparison can occur because of the complexity of the terrain. However, we cannot yet identify the reason with certainty using the data and CLM. This examination can be a simple comparison between model and data but also a test for the measurements. We also found a partial solution through a multi-layered scheme (CLM-ml), which is too complex to describe in this manuscript, but is the subject of one we will soon submit.

[Major Comment 1] (a) ET/TR: If I interpret Fig. 6 c correctly, the average TR from the sapflow measurements is as large or even larger than the average ET from the EC measurements (integrated over the entire day). This would imply that either the sapflow measurements overestimate TR (because the sampled trees are not representative? The setup described in Aparecido et al., 2016 is convincing though.) or the EC measurements underestimate ET (because the EC method is problematic on sloped terrain?), as one would expect that ET is higher than TR at a site with considerable interception by leaves. In fact, the simulated TR of the CLM versions seems to be quite realistic. (b) Also, I wonder whether it makes sense to exclude nighttime water fluxes from the analysis with the argument that CLM does not represent nighttime TR. As the authors mention in lines 240-245, the sapflow measurements exhibit a temporal lag, where part of the daytime TR originates from plant water uptake during the night. Wouldn't it make more

sense to compare values integrated over the entire day for a fairer comparison? I agree that the diurnal cycle of ET is relevant and should ideally be captured by the model. But still as a starting point, a good representation of the daily average value is already important.

**Response to Major Comment 1(a):** This is a good point that the transpiration rate (TR) seems higher than the total vapor flux from the eddy-covariance (EC) measurement, which is one of the interesting parts of our site and still under on-going study. Since TR was estimated by many sensors, and the eddy-covariance system was placed above the canopy and the sensor was located at a quite open place concerning the predominant wind direction, we assumed that those observations were reasonable. Here, the most likely reason of this issue is a spatial discrepancy between the footprint of the EC system and the extent of the sapflow plot. Our previous work has shown that large trees contribute disproportionately to the overall flux. In addition to the scale issue between two methods, we can also suspect the influence of a near emergent tree on EC measurement. This interference by the up-slope tree can occur anywhere in a mountain area. Therefore, it is still an interesting question of how to handle the horizontal influence beyond the traditional turbulence model. We have also suggested a partial solution to this issue in a follow-on manuscript, which uses a multi-layer canopy model (CLM-ml). We added a paragraph to discuss this:

1. L506 - "From the similarity of two observations (EC vs. TR), we suspect the influence of a near emergent tree on the EC measurements, which is possibly diagnosed by the advanced model (e.g., profiled simulation). Such interference by the up-slope tree can occur anywhere in a sloped area and the CLM insufficiently represents spatial variability. Also, the TR was estimated using more than 40 trees with a 2200 $m^2$ plot. However, this plot is not necessarily situated such that. In this case, a demographic model for TR and a multi-layer model for EC measurements may be useful to give more perspectives and address this problem. These might resolve the spatial scale issue and provide a method to handle some heterogeneity in the canopy (e.g., the emergent tree) beyond the traditional turbulence model."

**Response to Major Comment 1(b):** Yes, the setup described in Aparecido (2016) is convincing, although the temporal-lag of the sap-flow rate cannot clearly explain yet. We suspect this is related to the water storage capacity (i.e., capacitance) of such large trees. Hence, we updated [Table 1] to provide daily TR with and without nighttime TR. The daytime diurnal variation for sap-flow showed an apparent delay that we can identify a particular time between two peaks compared to the EC data. We agree with the reviewer that the nighttime sap-flow rate can occur to recharge the sap water because there are many tall and huge trees. The nighttime sap-flow is a quite constant flow (long-delayed flow) [Figure R1]. However, Figure 6d shows the sap-flow rate still reasonably follows the simulated TR on a short time scale. This was why we removed night-time TR in this analysis with the assumption that it is not realistic. Therefore, we cannot confidently conclude that the sap-flow data has such a long delay because the simulated transpiration in CLM immediately responds to sun-light. We need to note that the sap-flow rate was measured through many trees. We believe that the error of the sap-flow is minimal to represent the forest, although the upscaling contains different temporal delays. We updated the manuscript based on this discussion as below:

2. L253 - "However, taking into account that the nighttime sap-flow rate possibly occurs to recharge the sap water stored with the tree boles, an additional comparison was made without the elimination of nighttime value."

[Figure]

*Figure R3: Daily TR and Daytime TR comparison.*

**[Major Comment 2]** Leaf wetness: I am not sure whether I understood the normalization correctly. 0 corresponds to complete dryness of the leaves and 1 for $f_{wet} = f_{wetmax}$ in the respective model configuration? If this is the case the actual maximum in the diurnal cycle of the leaf wetness in CLM5 would be ~0.7·0.05 = 0.035. This would mean that CLM5 vastly underestimates the leaf wetness compared to the measurements rather than overestimate it as Fig. 7 suggests. Also why did you not test for an intermediate $f_{wetmax}$ (e.g., 0.5)?

**Response to Major Comment 2:** Yes, this can be confusing. The reviewer understood correctly about leaf wetness ($f_{wet}$), which ranges from 0 to 1 in CLM4.5. However, CLM5 forces the use of a range from 0 to 0.05, and 0.05 is not a scale factor. So, if $(W_{can} \cdot W_{max}^{-1})^{2 \cdot 3^{-1}}$ was 0.7 in CLM4.5, it would be 0.05 in CLM5. It is just giving a limit on the leaf wetness that cannot exceed 0.05. Another confusing part is the canopy water amount ($W_{can}$) at Eq (4). This amount was not related to 0.05 but limited by $W_{max}$. In summary, $f_{wet}$ was used for the evaporation rate and transpiration rate, not for the amount. With $f_{wetmax}$ =0.05, $W_{can}$ tends to hold more water due to the low evaporation rate. Therefore, for Figure 7b, this leaf wetness was re-estimated using $W_{can}$ for a fair comparison, as described at line 275. About $f_{wetmax} = 0.5$, this is not complicated process so we can expect that $f_{wetmax} = 0.5$ will give a result between $f_{wetmax} = 0.05$ and $f_{wetmax} = 1$. To clarify, we added the following text:

   3.  L182 - "For instance, if $f_{wet}$ was 0.7, $f_{wet}$ would become 0.05".

**[Specific Comments]:**

**S1.** L17: I am not sure what the authors mean by climate cycles.
**Response:** L17: "climate" was replaced by "carbon"

**S2.** L24: Greater energy exchange than what?
**Response:** L24: replaced with "greater energy exchange than a temperate forest"

**S3**. L105: A brief statement about the seasonality could be of interest to the reader here.

**Response:** L110: added "The dry season starts from January and continues until April, and the mean rainfall is about 195 mm per month. The wet season is from May until the end of the year: the average rainfall in the wet season is approximately 470 mm per month (Teale et al., 2014; Aparecido et al., 2016)."

**S4.** L108: The base of which mountain. Providing a map with the location of the two sites might help the reader to get a clearer picture of the field sites.
**Response:** A new Figure 1 was added. Also added "It shares a boundary with the Children's Eternal Rainforest" at L106.

**S5.** L131: 33 or 34 m? In Fig. 1 the EC is located at a height of 33 m. Also what does IRGA stand for?
**Response:** L138: updated them as "infrared gas analyzer (IRGA) are located at 33 m height"

**S6.** L158-160: Soil moisture could still limit stomatal conductance in the model. However, it is probably a fine choice to neglect soil moisture limitation in this study, as ET and the carbon fluxes are on the high side in the model.
**Response:** L164: "Soil moisture does not appear to limit stomatal conductance in the model; the predicted average value of transpiration wetness factor in CLM was typically around 95% in this study period and never fell below 50%for any 30-minute time period."

**S7-8.** L260: Are you sure you are talking of the canopy air temperature here and not the 2 m temperature? The canopy temperature Ts has a different definition if I understand correctly (see eq. 5.93 in the technical documentation of CLM5.0).
**Response:** Ts is canopy temperature, but CLM does not provide the value as an output. We used 2-m temperature as Ts. Actual Ts should be between 2-m air temperature and vegetation temperature. Based on [Figure 11d], we can identify that their values would be close to each other so that using 2 m temperature would not a problem at this site. We updated the manuscript as below:
L271: "CLM uses Ts term like Figure 3 as canopy Ta but not provides Ts as an output variable. This 2 m temperature, named as TSA in CLM, would nearest value from the canopy. Moreover, our profile data indicates that air temperature does not vary much in different height near the top canopy, and 28.075m is still within the canopy."

**S9.** L286-287: Albedo cannot explain the difference in the nighttime net radiation. Also, how is the albedo estimated?
**Response:** Albedo, which is calculated based on shortwave radiation, is not estimated at night and is simply set to one in CLM. Nighttime net radiation should come from the vegetation cooling (longwave radiation). Added "daytime" for net radiation at L300.

**S10.** L307: To me it is unclear what is meant by the BB parameter.
Response: L326: it is the Ball-Berry Model (BB) slope parameter. Added "slope" to clarify.

**S11.** Fig. 4: Are the APAR values in panels c and d from the observations or the model? Part of the discrepancy in the alphas could also originate from differences in PAR (Fig. 3 f).
**Response:** APAR was estimated based on the two-stream approximation in CLM using observed incoming solar radiation. This is the similar comments to Reviewer1, and we added some information about APAR estimation and some discussion about the possible error in estimating alphas:
L332: "The APAR, including sunlit and shade leaf area, was estimated in CLM using measured incoming solar radiation above the canopy at 44 m."

L340: "Of course, this analysis itself has a possible error by the eddy-covariance measurement and APAR estimation. APAR, which estimated by CLM, contains only sunlit and shade leaf area, making it too simplistic. Also, this method itself has a possible error, as shown in [Figure 4e; Figure4f]."

**S12.** L344-345: The $R^2$ increased marginally for CLM5 when introducing phi=0.25. But, the slope and intercept are clearly deteriorated by this modification. So, I wouldn't really talk of an improvement here.
**Response:** That is true; we clarify:
L366: "However, we cannot conclude that it was improved, since the low Phi changed the slope and intercept values. This change might also be influenced by other components such as leaf wetness."

**S13.** L390: Another relevant process could be energy storage in particular in the stem, which is missing in the default version of CLM5. This process was found to decrease the diurnal temperature range in forests and decrease the overestimation of turbulent heat fluxes in CLM5 (Swenson et al., 2019, Meier et al., 2019).
**Response:** A reviewer provided an excellent suggestion. We added such a discussion about energy storage:
L565: "Additionally, adding storage flux can be influential in a rainforest due to its dense and tall canopy (Heidkamp et al., 2018), and this storage flux is not represented in CLM. The heat storage can be related to air under the canopy, but the role of vegetation biomass is also significant. Considering the heat storage of vegetation biomass reduced the diurnal temperature range in other studies (Swenson et al., 2019; Meier et al., 2019)."

**S14.** L433: A nice study supporting and concretizing your claim for accounting for terrain effects: Fan et al. (2019).
**Response**: L72: we added a paragraph based on the reviewer's suggestion. It is as follows: "Land surface models have gradually increased in resolution with the improvement of observations through remote-sensing technology. These changes have highlighted the importance of spatial variability in the land surface system. However, the models still cannot fully reflect the complexity of the surface. The current oversimplified parameterization is one cause of model error (Singh et al., 2015; Wood et al., 2011). For instance, hydrological processes are well studied at the catchment scale and reflect topographic gradients, but LSMs are known to simplify the effect of the topographic slope (Fan et al., 2019a). Critical zone science has a gap from the Earth system model which normally focuses on vertical flow (Fan et al., 2019a; Clark et al., 2015). The failure to reflect spatial heterogeneity and hydrologic connectivity between large scale process (land-atmosphere fluxes) and microscale process (biogeochemical interactions) can be problematic (Clark et al., 2015)."

**[Technical Comments]:**
Response: We accept all these copy-editing changes as written, including the labeling of figures. For brevity, we do not list them here individually, but the new version of the manuscript reflects them.

**[Reference]**
Aparecido, L. M. T., Miller, G. R., Cahill, A. T., and Moore, G.W.: Comparison of tree transpiration under wet and dry canopy conditions in a Costa Rican premontane tropical forest, Hydrological Processes, 30, 5000–5011, https://doi.org/10.1002/hyp.10960, 2016.

Clark, M. P., Fan, Y., Lawrence, D. M., Adam, J. C., Bolster, D., Gochis, D. J., Hooper, R. P., Kumar, M., Leung, L. R., Mackay, D. S., Maxwell, R. M., Shen, C., Swenson, S. C., and Zeng, X.: Improving the

representation of hydrologic processes in Earth System Models, Water Resources Research, 51, 5929–5956, https://doi.org/10.1002/2015WR017096, 2015.

Fan, Y., Clark, M., Lawrence, D. M., Swenson, S., Band, L. E., Brantley, S. L., Brooks, P. D., Dietrich,W. E., Flores, A., Grant, G., Kirchner, J. W., Mackay, D. S., McDonnell, J. J., Milly, P. C. D., Sullivan, P. L., Tague, C., Ajami, H., Chaney, N., Hartmann, A., Hazenberg, P., 720 McNamara, J., Pelletier, J., Perket, J., Rouholahnejad-Freund, E., Wagener, T., Zeng, X., Beighley, E., Buzan, J., Huang, M., Livneh, B., Mohanty, B. P., Nijssen, B., Safeeq, M., Shen, C., van Verseveld, W., Volk, J., and Yamazaki, D.: Hillslope Hydrology in Global Change Research and Earth System Modeling, Water Resources Research, 55, 1737–1772, https://doi.org/10.1029/2018WR023903, 2019a.

Heidkamp, M., Chlond, A., and Ament, F.: Closing the energy balance using a canopy heat capacity and storage concept – a physically based approach for the land component JSBACHv3.11, 11, 3465–3479, https://doi.org/10.5194/gmd-11-3465-2018, 2018.

Meier, R., Davin, E. L., Swenson, S. C., Lawrence, D. M., and Schwaab, J.: Corrigendum: Biomass heat storage dampens diurnal temperature variations in forests, Environmental Research Letters, 14, 119 502, https://doi.org/10.1088/1748-9326/ab4a42, 2019.

Singh, R. S., Reager, J. T., Miller, N. L., and Famiglietti, J. S.: Toward hyper-resolution land-surface modeling: The effects of finescale topography and soil texture on CLM4.0 simulations over the Southwestern U.S., Water Resources Research, 51, 2648–2667, https://doi.org/10.1002/2014WR015686, 2015.

Swenson, S. C., Burns, S. P., and Lawrence, D. M.: The Impact of Biomass Heat Storage on the Canopy Energy Balance and Atmospheric Stability in the Community Land Model, Journal of Advances in Modeling Earth Systems, 11, 83–98, https://doi.org/10.1029/2018MS001476, 2019.

Teale, N. G., Mahan, H., Bleakney, S., Berger, A., Shibley, N., Frauenfeld, O.W., Quiring, S. M., Rapp, A. D., Roark, E. B., and Washington-Allen, R.: Impacts of vegetation and precipitation on throughfall heterogeneity in a tropical pre-montane transitional cloud forest, Biotropica, 46, 667–676, https://doi.org/10.1111/btp.12166, 2014.

Wood, E. F., Roundy, J. K., Troy, T. J., van Beek, L. P. H., Bierkens, M. F. P., Blyth, E., de Roo, A., Döll, P., Ek, M., Famiglietti, J., Gochis, D., van de Giesen, N., Houser, P., Jaffé, P. R., Kollet, S., Lehner, B., Lettenmaier, D. P., Peters-Lidard, C., Sivapalan, M., Sheffield, J., Wade, A., and Whitehead, P.: Hyperresolution global land surface modeling: Meeting a grand challenge for monitoring Earth's terrestrial water, Water Resources Research, 47, https://doi.org/10.1029/2010WR010090, 2011.